# Sterile-neutrino search based on 259 days of KATRIN data

The KATRIN Collaboration*

Neutrinos are the most abundant fundamental matter particles in the Universe and play a crucial part in particle physics and cosmology. Neutrino oscillation, discovered about 25 years ago, shows that the three known species mix with each other. Anomalous results from reactor and radioactive-source experiments[1] suggest a possible fourth neutrino state, the sterile neutrino, which does not interact through the weak force. The Karlsruhe Tritium Neutrino (KATRIN) experiment[2], primarily designed to measure the neutrino mass using tritium β-decay, also searches for sterile neutrinos suggested by these anomalies. A sterile-neutrino signal would appear as a distortion in the β-decay energy spectrum, characterized by a discontinuity in curvature (kink) related to the sterile-neutrino mass. This signature, which depends only on the shape of the spectrum rather than its absolute normalization, offers a robust, complementary approach to reactor experiments. Here we report the analysis of the energy spectrum of 36 million tritium β-decay electrons recorded in 259 measurement days within the last 40 eV below the endpoint. The results exclude a substantial part of the parameter space suggested by the gallium anomaly and challenge the Neutrino-4 claim. Together with other neutrino-disappearance experiments, KATRIN probes sterile-to-active mass splittings from a fraction of an eV$^2$ to several hundred eV$^2$, excluding light sterile neutrinos with mixing angles above a few per cent.

The three known neutrino flavours, electron ($\nu_e$), muon ($\nu_\mu$) and tau ($\nu_\tau$), are neutral elementary particles (leptons) that interact only through the weak force, making them challenging to detect and crucial for understanding both particle physics and cosmology[3]. Neutrinos produced in a specific flavour state can transform into other flavour states. This phenomenon, known as neutrino flavour oscillation, links flavour states ($\nu_e$, $\nu_\mu$, $\nu_\tau$), which determine their weak interactions, with mass states ($\nu_1$, $\nu_2$, $\nu_3$), which are crucial for understanding the dynamics of the Universe. A hypothetical fourth neutrino state ($\nu_4$), with a well-defined mass $m_4$, could exist as a natural extension of the Standard Model of particle physics[4]. This mass state would predominantly lack a marked flavour component, with small contributions from e, μ and τ flavours. This minimal flavour association is why it is often referred to as a 'sterile neutrino' ($\nu_s$). Active neutrinos can oscillate into sterile neutrinos, and mixing between these states acts as the fundamental mechanism for indirectly detecting sterile neutrinos. In the context of a 3 + 1 neutrino model[1], the extended PMNS matrix $U$ is a 4 × 4 unitary matrix that describes the mixing between flavour and mass eigenstates. The element $|U_{e4}|^2$, also labelled $\sin^2(\theta_{ee})$, sets the oscillation amplitude[3]. For direct comparison with disappearance oscillation experiments, this is often recast as $\sin^2(2\theta_{ee}) = 4\sin^2(\theta_{ee})(1 - \sin^2(\theta_{ee}))$. A sterile-neutrino signature typically manifests as deviations in the outgoing lepton flux or energy spectrum.

Interest in sterile neutrinos has been driven by a series of experimental anomalies reported over the past three decades. Among these, the gallium anomaly[5] (GA) and the reactor antineutrino anomaly[1] (RAA) are the most relevant for the work presented here. The GA refers to a deficit in electron neutrino flux observed during radiochemical experiments, which involve MeV-scale neutrinos and very short baselines of a few metres. Initially reported by the Gallium Experiment (GALLEX)[6] and the Soviet–American Gallium Experiment (SAGE) experiments[7,8], it was recently confirmed by the Baksan Experiment on Sterile Transitions (BEST) experiment[9,10]. By contrast, the RAA describes discrepancies between predicted and measured fluxes at longer baselines of about 10–100 m, with neutrino energies around 4 MeV. Several experiments, including DANSS (Detector of the reactor AntiNeutrino based on Solid Scintillator)[11], PROSPECT (Precision Reactor Oscillation and Spectrum Experiment)[12] and STEREO (Search for Sterile Reactor Neutrino Oscillations)[13] have reported null results in direct tests of sterile neutrino oscillations by searching for spectral distortions. By contrast, the Neutrino-4 experiment[14] has claimed a positive signal consistent with both the RAA and GA, reporting an oscillation pattern in the neutrino energy spectrum compatible with sterile neutrinos, with parameters $m_4 \approx 2.70 \pm 0.22$ eV and $\sin^2(2\theta_{ee}) = 0.36 \pm 0.12$. However, this result remains unconfirmed, and no scientific consensus has been reached[4]. In light of these conflicting findings, leading interpretations of the RAA increasingly point to biases in flux predictions or underestimated systematic uncertainties[1,15].

Other notable anomalies include observations from the Liquid Scintillator Neutrino Detector (LSND) experiment[16], which ran during the 1990s and detected an excess of electron antineutrino events in a muon antineutrino beam. This finding was interpreted as potential evidence for sterile-neutrino involvement. Around the same time, the KARMEN experiment[17], which tested the same oscillation channel, observed

no such excess, placing it in tension with LSND. Later, the MiniBooNE experiment[18–20], operating at approximately 10 times the energy and baseline of LSND, observed an unexpected surplus of electron neutrino and antineutrino events in a muon-flavour neutrino beam. These results further highlight the unresolved questions surrounding sterile neutrinos[21].

Despite variations in neutrino flavour, energy and baseline, these anomalies collectively hint at the possibility of non-standard neutrino oscillations involving sterile neutrinos associated with a mass range of 0.1 eV to several tens of eV (ref. 4). This is inferred from the relation $\Delta m_4^2 \approx \frac{1.27 \times E(\text{MeV})}{L(\text{m})}$ eV$^2$, linking the oscillation length to the energy-to-baseline ratio. However, their statistical significance, ranging from 2 to 4 standard deviations, offers only weak evidence.

The existence of sterile neutrinos remains controversial, primarily because of the challenges in fully understanding systematic uncertainties and backgrounds of each experiment. This complexity is reflected in a diverse experimental landscape. For instance, measurements sensitive to the same mixing channel as Karlsruhe Tritium Neutrino (KATRIN) ($|U_{e4}|^2$), such as Double Chooz[22] and Daya Bay[23], have reported null results. Moreover, experiments probing complementary channels, such as NOvA[24], MINOS/MINOS+ (Main Injector Neutrino Oscillation Search)[25] and Super-Kamiokande/T2K[26], which primarily constrain $|U_{\mu4}|^2$, have found no evidence for sterile neutrino mixing. Similarly, IceCube/DeepCore[27], sensitive to both $|U_{\mu4}|^2$ and $|U_{\tau4}|^2$, has also reported null results. Collectively, these findings reinforce the robustness of the three-flavour neutrino framework.

Moreover, although the involvement of sterile neutrinos cannot be ruled out, global fits have quantified significant tensions among the various experimental results[28]. For instance, the gallium anomaly points to larger mixing angles than those inferred from reactor data, and Neutrino-4 reports a signal not corroborated by other reactor experiments. These inconsistencies suggest that, if sterile neutrinos are responsible, scenarios beyond the minimal 3 + 1 active-sterile mixing framework may be required[29].

Taken together, these experimental results underscore the importance of continued precision measurements, such as those performed by KATRIN, to further probe the sterile neutrino hypothesis and clarify the origin of these anomalies.

## Sterile-neutrino search with KATRIN

The KATRIN experiment[2,30] aims primarily to determine the absolute neutrino mass scale by analysing the β-decay of molecular tritium.

$$\text{T}_2 \rightarrow {}^3\text{HeT}^+ + \text{e}^- + \bar{\nu}_\text{e}. \tag{1}$$

Towards this goal, KATRIN performs a precision measurement of the electron energy spectrum down to 40 eV below the endpoint at 18.57 keV. To date, KATRIN has set an upper limit on the effective neutrino mass at 0.45 eV (90% confidence level (CL)) and aims to achieve a sensitivity of less than 0.3 eV by the end of data collection in 2025 (ref. 31). KATRIN also enables a highly sensitive search for a potential fourth neutrino-mass state[32,33].

In the case of three active neutrinos, the differential spectrum $R_\beta(E, E_0, m_\nu^2)$ of the super-allowed tritium β-decay can be calculated with very high precision for a given value of electron kinetic energy $E$, the endpoint energy of the spectrum $E_0$ and the effective neutrino mass $m_\nu = \sqrt{\sum_{i=1}^3 |U_{ei}|^2 m_i^2}$. Its shape is determined by energy and momentum conservation, the Fermi function with $Z' = 2$ for the helium nucleus and calculable corrections related to rotational, vibrational and electronic excitations of the $\text{T}_2$ and $^3\text{HeT}^+$ (refs. 34–36).

During β-decay, a hypothetical neutrino mass eigenstate with mass $m_4$ may be emitted, with a probability determined by the mixing between electron neutrino and sterile neutrino states. To incorporate a sterile neutrino, the original model is extended to include an additional contribution associated with a fourth neutrino-mass state, $m_4^2$, and active-to-sterile mixing, $\sin^2(\theta_{ee})$. The resulting differential decay spectrum is modelled as

$$\begin{aligned} R_\beta(E, E_0, m_\nu^2, m_4^2, \theta_{ee}) &= \cos^2(\theta_{ee}) \cdot R_\beta(E, E_0, m_\nu^2) \\ &\quad + \sin^2(\theta_{ee}) \cdot R_\beta(E, E_0, m_4^2). \end{aligned} \tag{2}$$

Observationally, a sterile neutrino would lead to a twofold imprint on the electron spectrum: a distinct kink signature and a broader global distortion. The kink feature allows determination of the fourth neutrino mass $m_4$, as it occurs at an energy equal to the endpoint energy minus $m_4$. The amplitude of the global distortion is proportional to the mixing strength between sterile and active neutrinos, parametrized by $\sin^2(\theta_{ee})$. These phenomena are shown in Fig. 1 (right), highlighting the distinctive signature of a sterile neutrino on a β-decay spectrum.

KATRIN exploits its sub-eV sensitivity to the neutrino mass[31,37] to search for sterile neutrinos at the eV scale, focusing on masses and mixing angles suggested by the RAA and GA. Here, the same apparatus and dataset as used in ref. 31 are used.

The search conducted by KATRIN complements oscillation-based methods that directly detect neutrinos. Instead of direct detection, KATRIN measures the effects of sterile neutrinos on the electron energy spectrum. The experimental signature is well defined, supported by precise spectral measurements and validated through meticulous calibration. Furthermore, the search benefits from a high signal-to-background ratio across most of the sterile-neutrino mass range investigated.

## Setup and dataset

The KATRIN setup[30], which spans over 70 m, consists of three main modules: a high-luminosity windowless gaseous tritium source (WGTS) of up to 100 GBq, a high-resolution magnetic adiabatic collimation and electrostatic (MAC-E) high-pass filter and an electron detector (Fig. 1). The WGTS provides a stable, high-purity tritium source used at 30 K and 80 K by continuously injecting gas at the centre of its 10-m-long beam tube. Tritium is then removed downstream using differential and cryogenic pumping, reducing pressure by 12 orders of magnitude to minimize background in the spectrometer. β-Decay electrons are guided through the beamline by strong magnetic fields of the order of several tesla. The MAC-E filter[38,39], comprising the main and pre-spectrometer, precisely analyses electron energies. The magnetic field in the main spectrometer gradually decreases to $B_{\text{ana}} \lesssim 6.3 \times 10^{-4}$, ensuring adiabatic collimation of electron momenta. The spectrometers transmit electrons with kinetic energy above the retarding energy $qU$. With a filter width smaller than 2.8 eV, KATRIN can precisely identify a potential kink-like structure associated with a fourth neutrino state consistent with the sterile-neutrino interpretation of the RAA and GA. Electrons passing through the filter are counted by the focal-plane detector (FPD), a silicon PIN diode segmented into 148 pixels for spatial resolution[40]. KATRIN acquires the integral beta spectrum by recording count rates at various set points of retarding energy, $qU_i$, with $qU_i$ spanning $E_0 - 40$ eV $\leq qU_i \leq E_0 + 135$ eV. The integrated spectrum rate $R_{\text{model}}$ is given by

$$\begin{aligned} R_{\text{model}}&(A_S, E_0, m_\nu^2, \sin^2(\theta_{ee}), m_4^2, qU_i) \\ &= \left[ A_S \int_{qU_i}^{E_0} R_\beta(E, E_0, m_\nu^2, \sin^2(\theta_{ee}), m_4^2) \right. \\ &\quad \left. \times f(E, qU_i)\text{d}E \right] + R_{\text{bg}}(qU_i), \end{aligned} \tag{3}$$

where the differential spectrum $R_\beta$ from equation (2) is convolved with the response function $f$, which primarily accounts for scattering in the

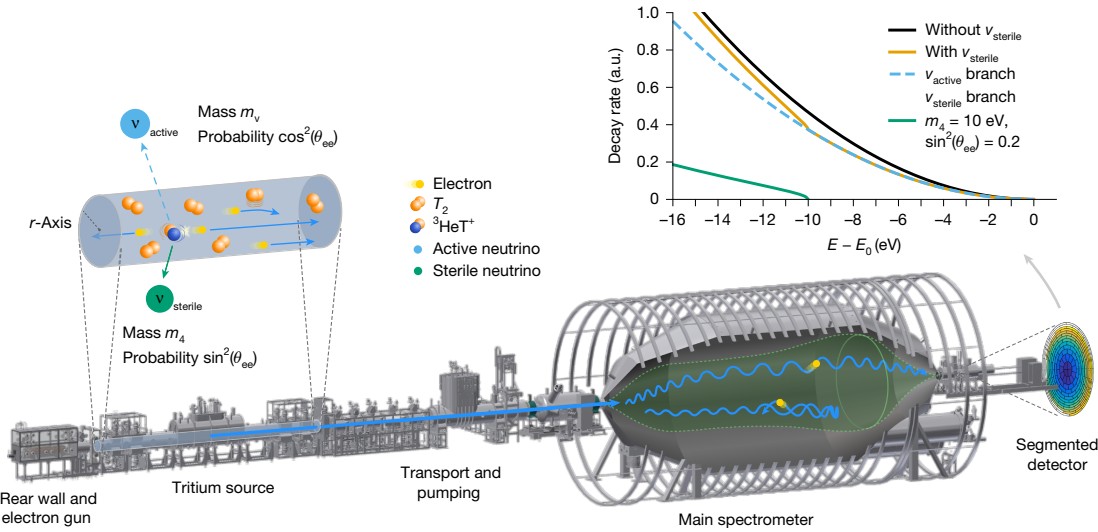

**Fig. 1 | The KATRIN beamline comprises six primary components, arranged from left to right.** The rear wall and electron gun, the windowless gaseous tritium source, the transport and pumping section, the pre- and main spectrometers, and the segmented focal-plane detector in which the electrons are recorded. The figure on the top right shows the expected spectral signature of a fourth neutrino state with mass $m_4 = 10$ eV in the tritium β-decay spectrum. The kink feature appears at $E_0 - m_4$, whereas the overall distortion is governed by the mixing amplitude $\sin^2(\theta_{ee})$. An exaggerated mixing amplitude is shown for clarity.

WGTS and the spectrometer transmission. A background rate $R_{bg}$ is also added to the tritium β-decay signal. This background consists of three components[31]: a dominant energy-independent background base $R_{bg}^{base}$, an unconfirmed but possible retarding energy-dependent background $R_{bg}^{qU}(qU_i)$ and a small contribution from electrons trapped in the Penning trap between the pre- and main spectrometers[41], denoted $R_{bg}^{Penning}(qU_i)$. The tritium signal normalization, $A_s$, is treated as a free-fit parameter. The endpoint energy, $E_0$, is left free, as it reflects the tritium $Q$-value and work function differences between the source, rear wall and spectrometer, introducing a sub-electronvolt uncertainty that prevents it from being fixed in the fit.

Data from the first[33] and second[32] measurement campaigns of KATRIN have already set stringent constraints on sterile-neutrino parameters, based on the analysis of 6 million electrons within the last 40 eV below the endpoint. An improved analysis, using data from the first five measurement campaigns conducted between March 2019 and June 2021, is presented in this study. Each scan included approximately 40 set points of the retarding energy $qU_i$ in the range of $[E_0 - 300$ eV, $E_0 + 135$ eV$]$ and lasted 2.1–3.3 h. Measurements above the endpoint enable accurate background determination. A total of 1,757 out of 1,895 scans were selected after data-quality cuts, comprising 36 million signal and background electrons within the same region of interest. This selection excludes scans in which monitoring systems indicated instabilities in key experimental parameters, such as electromagnetic fields or source conditions. This dataset allows a search for sterile neutrinos with a mass $m_4$ of up to approximately 33 eV.

During the first campaign (KNM1, for KATRIN Neutrino-mass Measurement 1), the tritium-gas column density was set below its design value at $\rho d = 1.08(1) \times 10^{21}$ m$^{-2}$, where $\rho$ represents the average gas density and $d = 10$ m is the length of the tritium source cavity. This value was subsequently increased to $\rho d = 4.20(4) \times 10^{21}$ m$^{-2}$ during the second campaign (KNM2). In the third campaign (KNM3), the background level was halved by moving the central analysing plane closer to the FPD detector, a configuration denoted as shifted analysing plane (SAP) that reduced volume-dependent backgrounds[42]. KNM3 was divided into its SAP setting (KNM3-SAP) and the symmetrical nominal analysing plane configuration (KNM3-NAP) for validation purposes. After KNM3, the calibration of the setup was enhanced by implementing a method that used $^{83m}$Kr conversion electrons to monitor electric potential variations within the tritium source. This approach involved co-circulating a tiny quantity of $^{83m}$Kr with tritium at a column density of $\rho d = 3.8 \times 10^{21}$ m$^{-2}$, which required raising the source temperature from 30 K to 79 K. In KNM4, measures were taken to address the scan-time-dependent background related to a Penning trap formed between the two spectrometers. This was achieved by lowering the pre-spectrometer potential, thereby eliminating the trap. Moreover, the scanning-time distribution for the scan steps was optimized to enhance neutrino-mass sensitivity, transitioning from the nominal KNM4-NOM to a more efficient configuration KNM4-OPT. Between KNM1 and KNM4, tritium accumulated on the gold surface of the rear wall, generating unwanted β-decay electrons. Before KNM5, a successful ozone cleaning was performed on the rear wall[43], effectively reducing the tritium activity by three orders of magnitude. Out of the 148 available pixels, 117 were used in KNM1 and KNM2, and 126 in KNM3-SAP, KNM3-NAP, KNM4-NOM, KNM4-OPT and KNM5. The remaining pixels were excluded because they either showed increased noise or were partially shadowed by structural components of the beamline.

## Testing the sterile-neutrino hypothesis

KATRIN measures the effective neutrino mass by analysing the shape of the tritium β-decay spectrum near the kinematic endpoint. In the standard neutrino mass analysis, four key parameters are fitted: the squared active-neutrino mass ($m_\nu^2$), the endpoint energy ($E_0$), the signal amplitude ($A_s$) and the background rate ($R_{bg}$). For the sterile-neutrino analysis, two additional parameters are introduced: the squared mass of the fourth neutrino state ($m_4^2$) and the mixing amplitude ($\sin^2(\theta_{ee})$). These are included in the model described in equation (3). Depending on the scenario being tested, the squared active-neutrino mass can either be left as a free parameter in the fit or set to zero. For the main results presented here, we assume a hierarchical scenario in which $m_{1,2,3} \ll m_4$, which allows $m_\nu$ to be set to zero. This is justified by neutrino oscillation data, which constrain the smallest possible effective neutrino mass to about 9 meV for the normal hierarchy ($m_1 < m_2 < m_3$) and 50 meV for the inverted hierarchy ($m_3 < m_1 < m_2$), well below the sensitivity of the current KATRIN measurements.

A systematic 50 × 50 logarithmic grid search spans $m_4^2$ from 0.1 eV$^2$ to 1,600 eV$^2$ and $\sin^2(\theta_{ee})$ from $10^{-3}$ to 0.5, examining parameter

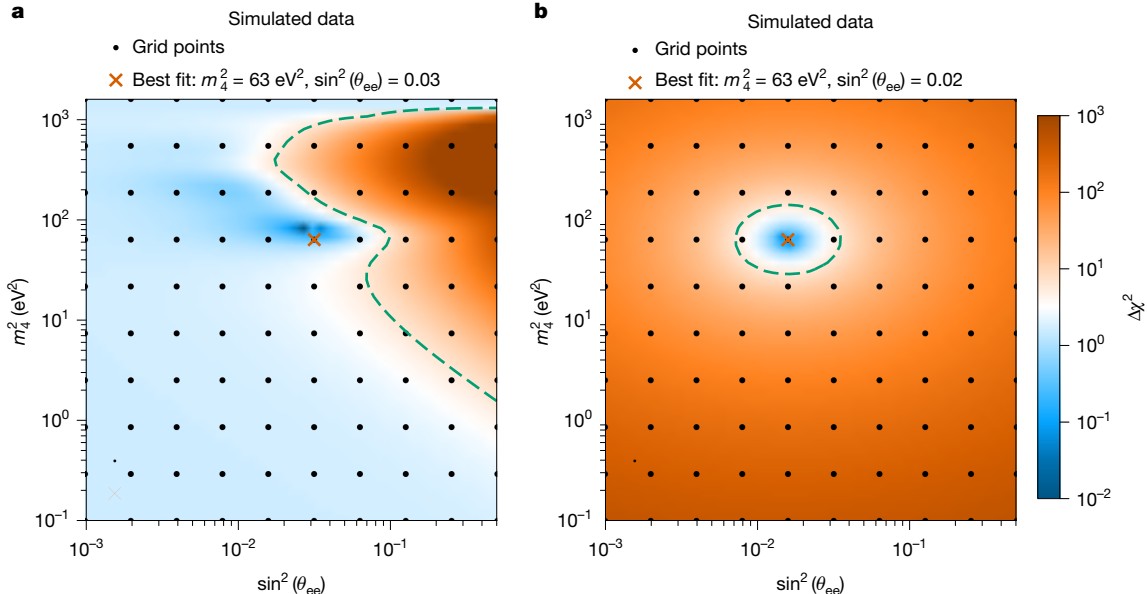

**Fig. 2 | Illustration of the grid-search analysis method based on Asimov datasets. a**, In the absence of a sterile-neutrino signal, showing an open contour as the dashed green line. **b**, In the presence of a sterile-neutrino signal, showing a closed contour as the dashed green line.

combinations and their impact on the spectral shape. At each grid point, the $\chi^2$ function is minimized with respect to three free parameters ($E_0$, $A_S$ and $R_{bg}^{base}$). To quantify the results, a confidence interval at 95% CL was derived based on Wilks' theorem[44] for two degrees of freedom, using $\Delta\chi^2 = \chi^2 - \chi_{min}^2 = 5.99$, where $\chi_{min}^2$ represents the minimum chi-square value across the grid. Systematic uncertainties were incorporated using the pull-term method, constrained by experimental data and calibration measurements.

As shown in Fig. 2, the analysis of simulated data produces an open contour in the parameter space when no sterile-neutrino signal is present (left), and a closed contour when such a signal is detected, indicating a statistically significant sterile neutrino presence (right). The resulting constraints on sterile-neutrino parameters are derived entirely from the shape of the experimental spectrum and validated through simulations.

To enhance the robustness of the analysis, the best fit $\chi_{min}^2$ is restricted to $m_4^2 < 1{,}000$ eV$^2$, ensuring that the sterile-neutrino branch, which appears at energies less than $E_0 - m_4$, spans multiple data points in the observed β-spectrum.

A blinding scheme (Methods) was applied to ensure an unbiased analysis.

Before analysing the actual datasets, Asimov datasets simulated under the respective operational conditions were thoroughly examined and cross-checked by two independent analysis teams.

Following these studies, the analysis of the actual data then proceeded, starting with independent assessments of the five campaigns (KNM1–KNM5) before unblinding and combining the datasets. During unblinding, the campaign-wise analysis showed an issue in KNM4, in which a distinct structure in the fit residuals was detected (Methods). A re-evaluation identified a problem with data combination, leading to the division of KNM4 into two sub-periods, KNM4-NOM and KNM4-OPT, for analysis. Moreover, some systematic effects were revisited and refined. This issue, discovered in the sterile-neutrino analysis, also affected the neutrino mass analysis[31].

## KATRIN in the light of neutrino-oscillation results

No significant sterile-neutrino signal was found in the KATRIN search. Consequently, we present the new 95% CL exclusion contour derived from the first five measurement campaigns (KNM1–KNM5), shown

in Fig. 3 (black). This limit is compared with constraints from other key experiments probing electron neutrino and antineutrino disappearance. As short-baseline neutrino oscillation experiments measure different observables than β-decay experiments, appropriate variable transformations are required for comparison. Although KATRIN directly probes $\sin^2(\theta_{ee})$, oscillation experiments typically report the effective mixing angle, defined as $\sin^2(2\theta_{ee}) = 4\sin^2(\theta_{ee})(1 - \sin^2(\theta_{ee}))$. The relevant mass splitting is approximated by $\Delta m_{41}^2 \approx m_4^2 - m_v^2$, valid to within approximately $2 \times 10^{-4}$ eV$^2$ (ref. 45). Figure 3 is obtained under the assumption $m_v^2 = 0$, so that the sterile-neutrino mass parameter corresponds directly to the squared mass splitting relative to the active state. Alternative treatments, including cases in which $m_v^2$ is left free, are discussed in the Methods.

For comparison, the exclusion contour from the first two KATRIN campaigns (KNM1 and KNM2) is also shown in light blue[32]. The improvements in KNM1–KNM5 reflect a sixfold increase in statistics and substantial improvements in the control of systematics.

The findings of KATRIN significantly constrain the parameter space associated with the RAA[1]. Although a region with small mixing angles remains viable, a large portion of the RAA parameter space is excluded. Furthermore, this KATRIN result challenges most of the parameter space favoured by the GA, recently reinforced by the BEST experiment[9,10]. In particular, the combined best-fit point from the BEST, GALLEX and SAGE experiments, at $\Delta m_{41}^2 \approx 1.25$ eV$^2$ and $\sin^2(2\theta_{ee}) \approx 0.34$, is excluded with 96.56% CL. Therefore, these results challenge the light sterile-neutrino hypothesis.

KATRIN results complement reactor-neutrino oscillation experiments by probing large $\Delta m_{41}^2$ values, reaching down to a few eV$^2$, whereas reactor experiments are more sensitive to lower $\Delta m_{41}^2$ values. The sensitivities of KATRIN and reactor-based experiments coincide at approximately $\Delta m_{41}^2 \approx 3$ eV$^2$ for a mixing angle of $\sin^2(2\theta_{ee}) \approx 0.1$. Among reactor experiments, PROSPECT (Fig. 3, yellow) currently provides the most stringent exclusion limits, benefiting from effective background suppression through pulse shape discrimination (PSD)[12]. STEREO[13], shown in light orange, is second in sensitivity, using a Gd-doped scintillator for systematic and background control. DANSS, represented in dark orange, achieves high sensitivity because of its large neutrino-candidate statistics, collecting approximately 4 million events over 3 years at various baselines[11].

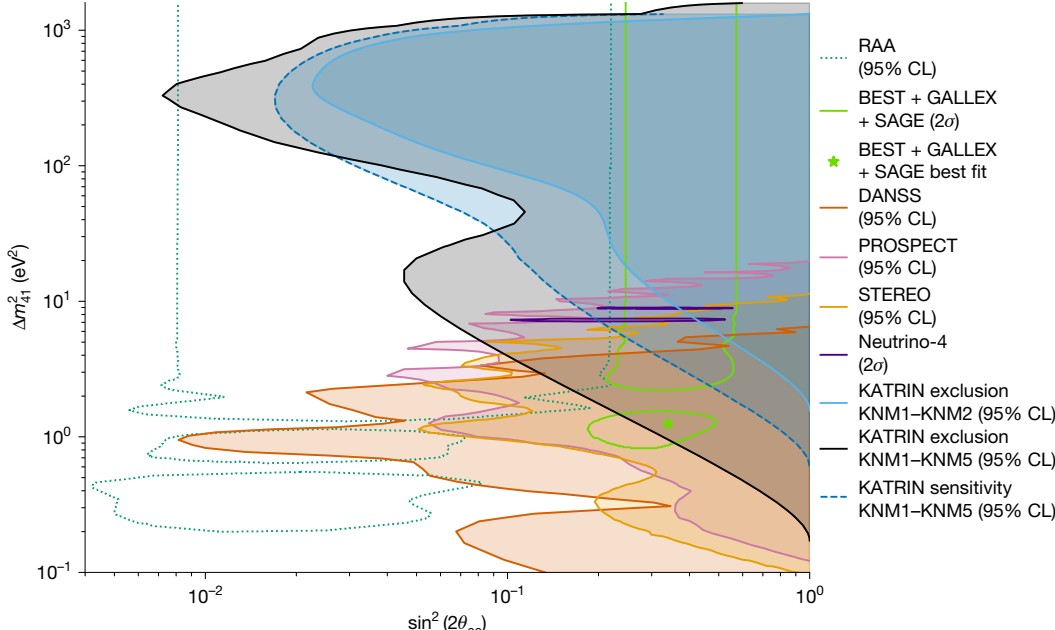

**Fig. 3 | The 95% CL exclusion curves in the ($\Delta m_{41}^2$, $\sin^2(2\theta_{ee})$) plane obtained from the analysis of the first five KATRIN campaigns with a fixed $m_v = 0$ eV (black).** The light and dark green contours denote the 3 + 1 neutrino oscillations allowed at the 95% CL by the RAA and GA[1,10]. The green star symbol represents the best-fit point from the BEST, GALLEX and SAGE experiments.

The first five measurement campaigns of KATRIN have fully excluded the 95% CL contour reported by Neutrino-4, which claimed oscillation evidence at $\Delta m_{41}^2 = 7.3$ eV$^2$ (ref. 14), corresponding to $m_4 \approx 2.70 \pm 0.22$ eV for $m_1 \ll m_4$. This claim has been a topic of debate, as the contours of Neutrino-4 reach only the sensitivity limits of PROSPECT and STEREO, in which their statistical power decreases. By contrast, the sterile neutrino parameter space favoured by Neutrino-4 is now fully excluded by KATRIN, which rejects the Neutrino-4 best-fit point at 99.99% CL.

## Conclusions and outlook

Using data from the first five measurement campaigns (KNM1–KNM5), conducted between 2019 and 2021 and comprising 36 million electrons near the tritium β-decay endpoint, KATRIN has carried out a new search for light sterile neutrinos. KATRIN places stringent constraints on much of the parameter space suggested by the RAA and GA, except for cases with very small mixing angles. KATRIN also disfavours the parameter region indicated by the Neutrino-4 claim[14]. The findings of KATRIN complement reactor-based oscillation experiments, such as PROSPECT[12] and STEREO[13], extending the sterile neutrino parameter space probed to larger mass-splitting scales.

With data collection continuing through 2025, the sensitivity of KATRIN is expected to increase substantially (Methods), allowing for improved searches for light sterile neutrinos. The dataset will then exceed 220 million electrons in the region of interest, more than six times the current statistics. In 2026, KATRIN will be upgraded with the TRISTAN detector[46], enabling differential measurements of the full tritium β-decay spectrum. TRISTAN will complement the current KATRIN setup, which focuses on the endpoint region, by probing the entire spectrum. This high-statistics approach will allow the search for keV-scale sterile neutrinos, potential dark matter candidates, with mixing angles as low as one part per million (ref. 47).

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

**The KATRIN Collaboration**

H. Acharya[1], M. Aker[2], D. Batzler[2], A. Beglarian[3], J. Beisenkötter[4], M. Biassoni[5], B. Bieringer[4], Y. Biondi[2], M. Böttcher[4], B. Bornschein[2], L. Bornschein[2], M. Carminati[6,7], A. Chatrabhuti[8], S. Chilingaryan[3], D. Díaz Barrero[2], B. A. Daniel[9], M. Descher[2], O. Dragoun[10], G. Drexlin[11], F. Edzards[12], K. Eitel[2], E. Ellinger[13], R. Engel[2], S. Enomoto[14], I. Fallböhmer[15], A. Felden[2], C. Fengler[2], C. Fiorini[6,7], J. A. Formaggio[16], C. Forstner[12], F. M. Fränkle[2], G. Gagliardi[5,17], K. Gauda[4], A. S. Gavin[1,18], W. Gil[2], F. Glück[2], R. Grössle[2], T. Höhn[2], K. Habib[2], V. Hannen[4], L. Haßelmann[2], K. Helbing[13], H. Henke[2], S. Heyns[2], R. Hiller[2], D. Hilleisheimer[2], D. Hinz[2], A. Jansen[2], C. Köhler[12,15], K. Khosonthongkee[19], J. Kohpeiß[2], L. Köllenberger[2], A. Kopmann[3], N. Kovač[2], L. La Cascio[11], L. Laschinger[12,15], T. Lasserre[12,15,20]✉, J. Lauer[2], T.-L. Le[2], O. Lebeda[10], B. Lehnert[21], A. Lokhov[11], M. Machatschek[2], A. Marsteller[2], E. L. Martin[1,18,22], K. McMichael[9,23], C. Melzer[2], L. E. Mettler[2], S. Mertens[12,15], S. Mohanty[2], I. Mostafa[3], I. Müller[2], A. Nava[5,17], H. Neumann[24], S. Niemes[2], I. Nutini[12,15], A. Onillon[12,15], D. S. Parno[9], M. Pavan[5,17], U. Pinsook[8], J. Plößner[12,15], A. W. P. Poon[21], J. M. L. Poyato[25], F. Priester[2], J. Ráliš[10], M. Röllig[2], S. Ramachandran[13], R. G. H. Robertson[14], C. Rodenbeck[2], R. Sack[2], A. Saenz[26], R. Salomon[4], J. Schürmann[4,26], P. Schäfer[2], A.-K. Schütz[21], M. Schlösser[2], L. Schlüter[21], S. Schneidewind[4], U. Schnurr[2], A. Schwemmer[12,15], A. Schwenck[2], M. Šefčík[10], D. Siegmann[12], F. Simon[3], J. Songwadhana[19], F. Spanier[27], D. Spreng[12], W. Sreethawong[19], M. Steidl[2], J. Štorek[2], X. Stribl[12,15], M. Sturm[2], N. Suwonjandee[8], N. T. Jerome[3], H. H. Telle[25], T. Thümmler[2], L. A. Thorne[28], N. Titov[29], I. Tkachev[29], K. Trost[2], K. Urban[12], D. Vénos[10], K. Valerius[2], S. Wüstling[3], C. Weinheimer[4], S. Welte[2], J. Wendel[2], C. Wiesinger[12,15], J. F. Wilkerson[1,18,30], J. Wolf[11], J. Wydra[2], W. Xu[16], S. Zadorozhny[29] & G. Zeller[2]

[1]Department of Physics and Astronomy, University of North Carolina, Chapel Hill, NC, USA. [2]Institute for Astroparticle Physics (IAP), Karlsruhe Institute of Technology (KIT), Eggenstein-Leopoldshafen, Germany. [3]Institute for Data Processing and Electronics (IPE), Karlsruhe Institute of Technology (KIT), Eggenstein-Leopoldshafen, Germany. [4]Institute for Nuclear Physics, University of Münster, Münster, Germany. [5]Sezione di Milano-Bicocca, Istituto Nazionale di Fisica Nucleare (INFN), Milano, Italy. [6]Dipartimento di Elettronica, Informazione e Bioingegneria, Politecnico di Milano, Milano, Italy. [7]Sezione di Milano, Istituto Nazionale di Fisica Nucleare (INFN), Milano, Italy. [8]Department of Physics, Faculty of Science, Chulalongkorn University, Bangkok, Thailand. [9]Department of Physics, Carnegie Mellon University, Pittsburgh, PA, USA. [10]Nuclear Physics Institute, Czech Academy of Sciences, Řež, Czech Republic. [11]Institute of Experimental Particle Physics (ETP), Karlsruhe Institute of Technology (KIT), Karlsruhe, Germany. [12]TUM School of Natural Sciences, Physics Department, Technical University of Munich, Garching, Germany. [13]Department of Physics, Faculty of Mathematics and Natural Sciences, University of Wuppertal, Wuppertal, Germany. [14]Center for Experimental Nuclear Physics and Astrophysics, and Department of Physics, University of Washington, Seattle, WA, USA. [15]Max-Planck-Institut für Kernphysik, Heidelberg, Germany. [16]Laboratory for Nuclear Science, Massachusetts Institute of Technology, Cambridge, MA, USA. [17]Dipartimento di Fisica, Università di Milano-Bicocca, Milano, Italy. [18]Triangle Universities Nuclear Laboratory, Durham, NC, USA. [19]School of Physics and Center of Excellence in High Energy Physics and Astrophysics, Suranaree University of Technology, Nakhon Ratchasima, Thailand. [20]DRF/IRFU, CEA, Université Paris-Saclay, Gif-sur-Yvette, France. [21]Nuclear Science Division, Lawrence Berkeley National Laboratory, Berkeley, CA, USA. [22]Department of Physics, Duke University, Durham, NC, USA. [23]Department of Physics, Washington and Jefferson College, Washington, PA, USA. [24]Institute for Technical Physics (ITEP), Karlsruhe Institute of Technology (KIT), Eggenstein-Leopoldshafen, Germany. [25]Departamento de Química Física Aplicada, Universidad Autonoma de Madrid, Madrid, Spain. [26]Institut für Physik, Humboldt-Universität zu Berlin, Berlin, Germany. [27]Institute for Theoretical Astrophysics, University of Heidelberg, Heidelberg, Germany. [28]Institut für Physik, Johannes-Gutenberg-Universität Mainz, Mainz, Germany. [29]Institute for Nuclear Research of the Russian Academy of Sciences, Moscow, Russia. [30]Oak Ridge National Laboratory, Oak Ridge, TN, USA. ✉e-mail: thierry.lasserre@mpi-hd.mpg.de

## Methods

### Experimental setup

The KATRIN experiment measures the electron energy spectrum of tritium β-decay near the kinetic endpoint $E_0 \approx 18.6$ keV. The 70-m-long setup comprises a high-activity gaseous tritium source, a high-resolution spectrometer using the MAC-E-filter principle and a silicon p-i-n diode detector[2,30].

Molecular tritium gas with high isotopic purity (up to 99%; ref. 48) is continuously injected into the middle of the WGTS, in which it streams freely to both sides. At the ends of the 10-m-long beam tube, more than 99% of the tritium gas is pumped out using differential pumping systems. With a throughput of up to 40 g day$^{-1}$, an activity close to 100 GBq can be achieved[49,50]. The β-electrons are guided adiabatically by a 2.5 T magnetic field through the WGTS[51]. Before entering the spectrometer, they traverse two chicanes, in which the residual tritium flow is reduced by more than 12 orders of magnitude through differential[52] and cryogenic pumping[53]. The β-electrons flowing in the upstream direction of the beamline reach the gold-plated rear wall, where they are absorbed. Besides separating the WGTS from the rear part of the KATRIN experiment that houses calibration tools, the rear wall also controls the source potential by a tunable voltage of $\mathcal{O}$(100 mV).

Electrons guided towards the detector are subjected to energy filtering by two spectrometers. A smaller pre-spectrometer first rejects low-energy β-particles. The precise energy selection with $\mathcal{O}$(1 eV) resolution is then performed by the 23-m-long and 10-m-wide main spectrometer. In both spectrometers, only electrons with a kinetic energy above the threshold energy $qU$ are transmitted (high-pass filter). The electron momenta $\mathbf{p} = \mathbf{p}_\perp + \mathbf{p}_\parallel$ are collimated adiabatically such that the transverse momentum is reduced to a minimum towards the axial filter direction by gradually decreasing the magnetic field towards the so-called analysing plane. In the main spectrometer, the magnetic field is reduced by four orders of magnitude to $B_{ana} \lesssim 6.3 \times 10^{-4}$ T. Behind the main spectrometer exit, the magnetic field is increased to its maximum value of $B_{max} = 4.2$ T, resulting in a maximum acceptance angle of $\theta_{max} = \arcsin(\sqrt{B_S/B_{max}}) \approx 51°$, where $\theta$ is the initial pitch angle of the electron. The filtered electrons are counted by the FPD, which is a silicon p-i-n diode segmented into 148 pixels (ref. 40). This detector is regularly calibrated with a $^{241}$Am source to ensure stable performance. Its efficiency is about 95%, with only small variations between pixels that remain constant over time. Effects that do not depend on the retarding potential are accounted for by the free normalization combined fit across groups or patches of pixels[54].

Background electrons are indistinguishable from tritium β-decay electrons and thus contribute to the overall count rate at the detector. There are different sources and mechanisms that generate the background events. The majority originates from the spectrometer section of the experiment. Secondary electrons are created by cosmic muons and ambient gamma radiation on the inner spectrometer surface[55,56] but are mitigated by magnetic shielding and a wire electrode system[30]. The decay of $^{219}$Rn and $^{220}$Rn inside the main spectrometer volume is reduced by cryogenic copper baffles that are installed in the pumping ducts of the main spectrometer[57]. Another source of background is electrons from radioactive decays produced in the low-magnetic-field part of the spectrometer[58]. These primary electrons can be trapped magnetically, ionizing residual-gas molecules and producing secondary electrons that are correlated in time, leading to a background component with a non-Poisson distribution. The dominating part of the background stems from the ionization of neutral atoms in highly excited states, which enter the main spectrometer volume in sputtering processes (decay of residual $^{210}$Pb) at the inner surface of the spectrometer. The low-energy electrons emitted in this process are accelerated to signal-electron energies and guided to the detector. The magnitude of this background component scales with the flux tube volume in the re-acceleration part of the main spectrometer. A re-configuration of the electromagnetic fields in the main spectrometer, called the SAP setting, reduces the background by a factor of 2 by shifting the plane of minimal magnetic field from the nominal position in the centre of the spectrometer towards the detector while compressing the flux tube at the same time[42,59]. After successful tests, this configuration was set as the new standard (see the next section). There is also the possibility of a Penning trap being formed between the pre- and main spectrometer. The stored particles can increase the background rate. To counter this effect, a conductive wire is inserted between scan steps into the beam tube to remove the stored particles[41]. Because the duration of the scan steps differs, this creates a scan-time-dependent background. Reducing the time between particle removal events and lowering the pre-spectrometer potential enabled a full mitigation of the scan-step-duration-dependent background starting with the KNM5 measurement period. Moreover, the transmission and detection probabilities of background electrons may slightly depend on the retarding-potential setting, potentially causing a retarding-potential-dependent background. This effect is addressed in the analysis through an additional slope component, constrained by dedicated background measurements[31].

### KNM12345 dataset

The data collected by KATRIN is organized into distinct measurement periods, referred to as KATRIN Neutrino Measurement (KNM) campaigns. The integral β-spectrum is measured through a sequence of defined retarding-energy set points, which we refer to as a scan. Each dataset comprises several hundred β-spectrum scans, with individual scan durations ranging from 125 min to 195 min. The measurement time distribution (MTD), shown in Extended Data Fig. 1 (bottom), determines the time allocated for each $qU_i$, optimized for maximizing sensitivity to a neutrino-mass signal, where the index $i$ corresponds to different retarding energy settings. The energy interval typically spans $E_0 - 300$ eV $\leq qU_i \leq E_0 + 135$ eV, following an increasing, decreasing or random sequence. The analysis presented in this work uses set points ranging from $E_0 - 40$ eV to $E_0 + 135$ eV and is based on the first five measurement campaigns (KNM1–KNM5), summarized in Extended Data Table 1. Extended Data Fig. 2 shows the electron energy spectra for each of the five individual campaigns.

The first measurement campaign of the KATRIN experiment, KNM1, started in April 2019 with a relatively low source-gas density of $\rho d = (1.08 \pm 0.01) \times 10^{21}$ m$^{-2}$ compared with the design value of $\rho d = 5 \times 10^{21}$ m$^{-2}$. Here, $\rho$ denotes the average density of the source gas, and $d = 10$ m is the length of the tritium source. The source was operated at a temperature of 30 K. The measurement in KNM1 lasted for 35 days, recording a total of about 2 million electrons. The sterile-neutrino analysis of this dataset was published in ref. 33.

After a maintenance break, the next campaign, KNM2, started in October of the same year and lasted for 45 days, only 10 days longer than KNM1. However, more than double the number of electrons was measured because of the increased source-gas density. The density was raised to $\rho d = (4.20 \pm 0.04) \times 10^{21}$ m$^{-2}$, which corresponds to 84% of the design value. Although the background rate was reduced from 0.29 count per second (cps) in KNM1 to 0.22 cps, it was still above the anticipated design value of 0.01 cps (ref. 2). The sterile-neutrino analysis of this dataset, in combination with KNM1, was published in ref. 32.

To further reduce the background rate, a new electromagnetic-field configuration[42] was tested in the next campaign, starting in June 2020. KNM3 was split into two parts to validate the new shifted-analysing-plane (SAP) setting in contrast to the nominal (NAP) setting. Before the start of the measurement, the source temperature was increased to 79 K. This allowed the co-circulation of gaseous krypton $^{83m}$Kr with the tritium gas to perform simultaneous calibration measurements and β-scans[60,61]. In the first part of KNM3, KNM3-SAP, the β-spectrum was measured in the SAP setting for 14 days, and the background was subsequently reduced to 0.12 cps. Although the SAP setting reduced the background almost by a factor of 2, the increased inhomogeneities in the magnetic and

electric fields require the segmentation of the detector analysis into 14 patches with 14 individual models[59]. In KNM3-NAP, it was demonstrated that switching between both settings works, and the background rate increased to 0.22 cps, as expected. The second part also lasted 14 days, and about 2.5 million electrons were measured in all of KNM3.

Before launching a new measurement campaign, a target source-gas density of $\rho d = 3.8 \times 10^{21}$ m$^{-2}$ was chosen to keep the same experimental conditions for future measurements. With the start of KNM4 in September 2020, one of the challenges was to reduce the background caused by charged particles accumulated in the Penning trap between the main and pre-spectrometer. The accumulation was time dependent, as the trap was emptied after each $qU_i$. To reduce this background, the run durations were shortened to empty the trap more often. Finally, the Penning-trap effect was eliminated by lowering the pre-spectrometer voltage. Furthermore, the MTD was optimized during the campaign to increase the neutrino-mass sensitivity. This sequence defines the split into the first part, KNM4-NOM, and the second part, with an optimized MTD, KNM4-OPT. In all of KNM4, more than 10 million electrons were recorded in 79 measurement days.

The final dataset used in this analysis is KNM5, which started in April 2021. It contains more than 16 million electrons recorded in 72 measurement days. Before the start of the measurement, the rear wall was cleaned of its residual tritium with ozone. This step was necessary to address the accumulation of tritium on the rear wall, which produced an additional spectrum on top of the tritium β-spectrum, introducing new systematic uncertainties[31].

## Model description

**General model.** The KATRIN experiment measures the neutrino mass by analysing the distortions near the endpoint of the tritium β-decay spectrum. The model used for this analysis is described in the main text and follows the formalism outlined in equations (2) and (3). It includes the theoretical β-spectrum, the experimental response function and parameters such as the active-neutrino mass squared $m_\nu^2$, sterile-neutrino mass squared $m_4^2$ and the mixing parameter $\sin^2(\theta_{ee})$. The simulated imprint of a fourth mass eigenstate, with a mass of 10 eV and mixing of 0.05, is shown in Extended Data Fig. 1. To address the computational challenge posed by 1,609 data points requiring $\mathcal{O}(10^3)$ evaluations of $R_{model}(A_S, E_0, m_\nu^2, m_4^2, \theta_{ee}, qU_i)$ per minimization step, an optimized direct-model calculation[62] and a neural-network-based fast model prediction[63] were used, as detailed in sections 'The KaFit analysis framework' and 'The Netrium analysis framework', respectively.

**The KaFit analysis framework.** KaFit (KATRIN Fitter) is a C++-based fitting tool developed for the KATRIN data analysis. It is part of KASPER— the KATRIN Analysis and Simulations Package[64]. KaFit uses the MIGRAD algorithm from the MINUIT numerical minimization library[65] to fit KATRIN data. The fit minimizes the likelihood function $-2\log(L)$ in equation (4) with respect to parameters of interest such as the squared active-neutrino mass $m_\nu^2$, $m_4^2$, $\sin^2(\theta_{ee})$ and nuisance parameters $\Theta$ (ref. 62).

Model evaluation is done by the Source Spectrum Calculation (SSC) module, which returns the predicted electron counts for a given set of input parameters. Within SSC, the integrated β-decay spectrum is calculated by numerically integrating the differential spectrum after convolving it with the experimental response function, as in equation (3). Using the measurement times and the experimental parameters, the SSC module computes the prediction for the number of counts for each retarding energy $qU_i$.

KaFit computes the negative logarithm of likelihood using the experimental and model electron counts. Twice the negative log-likelihood is referred to as $\chi^2$ in equation (4). KaFit minimizes the $\chi^2$ and returns the best-fit parameters and the corresponding $\chi^2_{min}$. A typical minimization requires $\mathcal{O}(10^4)$ evaluations of the $\chi^2$ function. For computational efficiency, KaFit and SSC make use of multiprocessor computing. The recalculation of all components of the spectrum model at each minimization step is avoided through caching, thereby reducing the overall minimization time by a factor of approximately $\mathcal{O}(10^3)$. Moreover, caching the response function $f(E, qU_i)$ (equation (3)) and reusing it across systematic studies further reduces redundant calculations, enhancing the overall efficiency. KaFit has been optimized to enhance the computational speed for sterile-neutrino analysis while maintaining sufficient precision for physics searches within the current statistical and systematic budget.

**The Netrium analysis framework.** In the second approach to perform the analysis, a software framework called Netrium[63] is used. In this case, the KATRIN physics model is approximated in a fast and highly accurate way using a neural network. It can predict the model spectrum $R_{model}(A_S, E_0, m_\nu^2, m_4^2, \theta_{ee}, qU_i)$ (equation (3)) for all main input parameters and improves the computational speed by about three orders of magnitude with respect to the numerical-model calculation. The neural network features an input and an output layer with two fully connected hidden layers in between. During the training process, $\mathcal{O}(10^6)$ sample spectra, calculated using the analytical model, are used to optimize the weights of the neural network.

**Comparison.** Cross-validation between the two different analysis approaches to calculate the model is a key component of the blinding strategy of KATRIN (see the next section) and serves as quality assurance for the analysis. In previous sterile-neutrino searches with the KATRIN experiment[32,33], the two described analysis approaches were benchmarked with the analysis software Simulation and Analysis with MATLAB for KATRIN (Samak)[66], not used for the current analysis. It is based on the covariance matrix approach and designed to perform high-level analyses of tritium β-spectra measured by the KATRIN experiment. For this comparison, the results of the combined sterile analysis, based on data from the first five measurement campaigns, are shown in Extended Data Fig. 3. The active-neutrino mass is set to $m_\nu^2 = 0$ eV$^2$. An excellent agreement is achieved for the exclusion contour and the best-fit parameter between the analyses conducted with KaFit and Netrium.

## Blinding strategy and unblinding process

KATRIN applies a rigorous blinding scheme to minimize human bias and ensure the integrity of the analysis process. This approach also enforces robust software validation to avoid potential programming errors. The benchmark for validation is established by fixing the active squared neutrino mass to $m_\nu^2 = 0$ eV$^2$ in all steps of the analysis.

The process began with the generation of twin Asimov datasets for each measurement campaign, in which all input values are set to their expected means[67]. These datasets were independently evaluated by two teams using the KaFit and Netrium software frameworks. Cross-validations confirmed an agreement within a few per cent of $\sin^2(\theta_{ee})$ between the results from the two teams. Only after this validation step was the combined analysis of all twin Asimov datasets performed. Once the validation using the Asimov data was completed, the real data were analysed on a campaign-by-campaign basis. Again, results from both teams were compared to ensure consistency. Only when the campaign-wise results demonstrated agreement did the analysis proceed to the final step, in which the complete, unblinded dataset was evaluated.

During the sterile-neutrino analysis of the KNM4 campaign, a closed contour with 99.98% CL was unexpectedly identified. This result was anomalous, given that all other campaigns, contributing the bulk of the statistics, consistently yielded open contours. The anomaly prompted a temporary suspension of the unblinding procedure and initiated a detailed technical investigation. The investigation uncovered a technical error in the combination of sub-campaigns within KNM4. Specifically, data from periods with different MTDs and effective endpoints cannot be directly stacked together. To address this, the dataset was

divided into two distinct periods: KNM4-NOM and KNM4-OPT, reflecting the respective sub-campaign configurations. Apart from correcting the data-combination process, the analysis framework was improved with updated inputs and enhanced validation checks to prevent similar issues in the future. The identification of this error during the sterile-neutrino search also led to a re-evaluation of the neutrino-mass analysis, which was based on the same dataset. Detailed descriptions of all modifications and their impact can be found in ref. 31.

### Results for individual campaigns

There are seven individual datasets (with KNM3 and KNM4 each split into two), corresponding to the first five measurement campaigns (KNM1–KNM5), which are described in detail in the section 'KNM12345 dataset'. Each of the datasets can be used separately to search for a sterile neutrino. For campaigns measured in the NAP setting, the counts of all active pixels are summed up to one spectrum. By contrast, datasets in the SAP setting feature a patch-wise data structure (14 patches), in which, for each patch, nine pixels with similar electromagnetic fields are grouped together. Therefore, each dataset in the SAP setting contains 14 different spectra.

A sterile-neutrino search using the grid-scan method is applied to the seven individual datasets, whose exclusion contours at 95% CL are shown in Extended Data Fig. 4. The differences in the accumulated signal statistics and their variations lead to different excluded regions in the sterile parameter space. The overall shape of the exclusion contours is very similar for all of the campaigns. The KNM4-NOM campaign enables the exclusion of a more extensive region of the sterile-neutrino parameter space associated with high sterile-neutrino masses; for low sterile-neutrino masses, the KNM5 campaign dominates.

### Results for combined campaigns

Across the seven campaigns, a total of 68,237 scan steps were recorded, collecting approximately 36 million counts within the analysis window. As described in the section 'KNM12345 dataset', data were grouped primarily into seven sets according to the experimental conditions and FPD pixel configurations. Within each dataset, electrons collected at a particular retarding potential were summed over all the scans. This is possible given the ppm-level ($10^{-6}$) high-voltage reproducibility of the main spectrometer[68]. For each of the NAP datasets (KNM1, KNM2 and KNM3-NAP), the counts from all the pixels were summed up to obtain datasets that feature counts per retarding potential. For the SAP datasets (KNM3-SAP, KNM4-NOM, KNM-OPT and KNM5), the detector pixels were grouped into 14 ring-like detector patches of nine pixels each, to account for variations in the electric potential and magnetic fields across the analysing plane[42]. Pixel grouping was determined based on transmission similarity. For each of the SAP datasets, counts per patch per retarding potential were obtained by summing over all scans and grouping pixels. These so-called merged datasets, containing 1,609 data points, were used in the sterile-neutrino analysis.

The combined data were analysed using the following definition of $\chi^2$:

$$
\begin{aligned}
\chi^2(\boldsymbol{\theta}) &= -2 \ln L(\boldsymbol{\theta}) \\
&= -2 \ln \prod_{i=0}^{I} f\left( N_{\mathrm{obs},i} \,\middle|\, N_{\mathrm{theo}}(qU_i, \boldsymbol{\theta}) \right) \\
&= -2 \sum_{i=0}^{I} \ln\left( f\left( N_{\mathrm{obs},i} \,\middle|\, N_{\mathrm{theo}}(qU_i, \boldsymbol{\theta}) \right) \right),
\end{aligned}
\tag{4}
$$

where $\boldsymbol{\theta}$ is the vector of fit parameters, $I$ denotes the total number of data points or retarding energy setpoints $qU_i$, $N_{\mathrm{obs},i}$ is the observed count at the $i$th setpoint, $N_{\mathrm{theo}}(qU_i, \boldsymbol{\theta})$ is the theoretically predicted count at the $i$th setpoint and $f(N_{\mathrm{obs},i}|N_{\mathrm{theo}}(qU_i, \boldsymbol{\theta}))$ represents the likelihood function, modelled using a Poisson distribution. The $\boldsymbol{\theta}$ parameter values were inferred by minimizing $\chi^2(\boldsymbol{\theta})$.

For the combined KNM1–KNM5 analysis, further merging of the datasets is not possible because of the significant differences in the experimental conditions in which they were measured. The combined $\chi^2$ function incorporates contributions from both Gaussian and Poissonian likelihoods. Negative log-likelihoods of Gaussian models ($\chi^2_{\mathrm{G}}$) were used for the NAP campaigns as the mean value of the counts summed over all pixels was large, allowing the counts to be modelled by a Gaussian distribution[31]. Negative log-likelihoods of Poissonian models ($\chi^2_{\mathrm{P}}$) were applied to the SAP campaigns, as the mean value of the counts summed over pixels in each patch was not large enough. The combined $\chi^2$ function is expressed as

$$
\begin{aligned}
\chi^2_{\mathrm{comb}}(\boldsymbol{\Theta}) = {}& \sum_{k \in \mathcal{I}_{\mathrm{G}}} \chi^2_{\mathrm{G},k}\left( \begin{matrix} m_{\mathrm{v}}^2,\ m_4^2,\ \sin^2\theta_{ee}, \\ E_{0k},\ \mathrm{Sig}_k,\ \mathrm{Bg}_k,\ \xi \end{matrix} \right) \\
&+ \sum_{l \in \mathcal{I}_{\mathrm{P}}} \sum_{p=0}^{13} \chi^2_{\mathrm{P},lp}\left( \begin{matrix} m_{\mathrm{v}}^2,\ m_4^2,\ \sin^2\theta_{ee}, \\ E_{0lp},\ \mathrm{Sig}_{lp},\ \mathrm{Bg}_{lp},\ \xi \end{matrix} \right) \\
&+ (\hat{\xi} - \xi)^{\mathrm{T}} \Sigma^{-1} (\hat{\xi} - \xi).
\end{aligned}
\tag{5}
$$

In this formulation, $\boldsymbol{\Theta}$ is the vector of the physics parameters and the nuisance parameters of all the campaigns. $\mathrm{Sig}_k$ and $\mathrm{Sig}_{lp}$ correspond to $A_{\mathrm{S}}$ in equation (3) and $\mathrm{Bg}_i$ and $\mathrm{Bg}_{ip}$ represent the energy-independent background rate $R_{\mathrm{bg}}^{\mathrm{base}}$. Correlations among nuisance parameters $\xi$ were captured by the covariance matrix $\Sigma$, with the mean values $\hat{\xi}$ determined from systematic measurements. In equation (5), $\mathcal{I}_{\mathrm{G}} = 1, 2, 3$-NAP represents the KNM-NAP campaigns modelled with Gaussian likelihoods, whereas $\mathcal{I}_{\mathrm{P}} = 3$-SAP, 4-NOM, 4-OPT, 5 corresponds to the KNM-SAP campaigns modelled with Poissonian likelihoods. The exclusion bounds in Extended Data Fig. 5 compare results from the first two campaigns (KNM1–KNM2) to all five campaigns (KNM1–KNM5). The analysis highlights how increased statistics significantly improve exclusion limits across all $m_4^2$ values.

### Systematic uncertainty

An accurate and detailed accounting of the systematic uncertainties is essential for obtaining robust estimates of $m_4^2$ and $\sin^2(\theta_{ee})$ from the tritium β-spectrum. Several systematic effects arise within the KATRIN beamline, influencing the shape of the measured spectrum. The systematic effects and inputs considered for this analysis are the same as in the neutrino-mass analysis[31] and are treated as nuisance parameters. The central values $\hat{\xi}$ and covariance $\Sigma$ of the systematics parameters were determined from calibration measurements and simulations. To include these effects in the likelihood function (equation (4)), they are modelled as a multivariate normal distribution $\mathcal{N}(\hat{\xi}, \Sigma)$. Subsequently, the likelihood function is multiplied by the normal distribution of each systematic effect. It decreases the likelihood function if the parameters deviate from their mean values $\hat{\xi}$.

To assess the impact of individual systematic uncertainties, a separate scan of the sterile parameter space is performed for each systematic effect, considering only the statistical uncertainty and the specific systematic uncertainty in each case. This is done on the simulated Asimov twin data with $m_{\mathrm{v}}^2 = 0$ eV$^2$. The impact of individual systematic uncertainties on the sensitivity contour is quite small. Therefore, to evaluate the influence of systematic uncertainties more quantitatively, a raster scan is performed. In this approach, for each fixed value of $m_4^2$ in the parameter grid, the $1\sigma$ sensitivity on the mixing $\sin^2(\theta_{ee})$ is calculated independently. By performing a raster scan, the number of degrees of freedom is effectively reduced to one at each grid point. The contribution of each systematic effect to the total uncertainty is determined using $\sigma_{\mathrm{syst}} = \sqrt{\sigma_{\mathrm{stat+syst}}^2 - \sigma_{\mathrm{stat}}^2}$. The raster scan results for the combined KNM1–KNM5 dataset are presented in Extended Data Fig. 6 for three values of $m_4^2$ as well as for all values of $m_4^2$ in Extended Data Fig. 7. We can see that the statistical uncertainty dominates over all systematic effects. Owing to increasing statistics at lower retarding

energies, the ratio of statistical to systematic uncertainties depends on the value of $m_4^2$. Source-related uncertainties dominate the overall systematic uncertainty. The largest impact stems from the uncertainty on the density of the molecular tritium gas in the source column. It is followed by the uncertainty of the energy-loss function, which describes the probability that electrons will lose a certain amount of energy when scattering with the molecules in the source. Depending on the value of $m_4^2$, variations in the source potential and the scan-time-dependent background also significantly affect the overall systematic uncertainty. The uncertainty from the scan-step-duration-dependent background is the first contribution of a non-source-related systematic effect.

### Final-state distribution systematics

One source of uncertainty affecting the shape of the β-electron spectrum in KATRIN arises from the final-state distribution (FSD) of molecular tritium. During the decay process, part of the available energy can excite the tritium molecule into rotational, vibrational (ro-vibrational) and electronic states, thereby reducing the energy transferred to the emitted electron. Consequently, the energy distribution of these states directly influences the differential β-decay spectrum used in the analysis.

In previous KATRIN campaigns, the systematic contribution of the FSD uncertainty was estimated by comparing two ab initio calculations—one from ref. 34 and an earlier version from ref. 69—leading to a rather conservative estimation. For the analysis presented here, the FSD-related uncertainty was re-evaluated following the procedure derived for the neutrino-mass analysis[36], which allows the estimation of the impact of FSD on the total uncertainty by considering the details of the calculation of the final-states distribution, such as the validity of theoretical approximations and corrections, the uncertainty of experimental inputs and physical constants, and the convergence of the calculation itself.

The same FSDs used in the neutrino-mass analysis, generated by perturbing the nominal set of input parameters entering the calculation, were used to create the twin Asimov datasets of the null hypothesis (no sterile neutrinos). Each dataset was then fitted with Netrium, trained on a single nominal FSD, with different values of $m_4$ and $\sin^2(\theta_{ee})$. By comparing the resulting sensitivity contours at the 95% CL ($\Delta\chi_{crit}^2 = 5.99$) with that from the nominal case, the impact of FSD uncertainties can be quantified.

This analysis was performed using the KNM5 campaign alone, due to differences between the models used to generate FSDs in different campaigns. Nonetheless, the result of this analysis, presented in Extended Data Fig. 8, demonstrates that the various FSD effects have a negligible impact on the sensitivity contours, justifying the use of the nominal FSD for fitting with sufficient accuracy.

### Applicability of Wilks' theorem

The contours and their associated confidence regions reported in earlier sections were based on the critical threshold values $\Delta\chi_{crit}^2$ corresponding to a desired CL. The sterile-neutrino search uses the $\Delta\chi^2$ test statistic:

$$\Delta\chi^2 = \chi^2(H_0) - \chi^2(H_1), \tag{6}$$

where $H_0$ is the assumed truth for the sterile-neutrino mass and mixing angle [$m_4^2$, $\sin^2(\theta_{ee})$], and $H_1$ represents the global best-fit hypothesis based on the observations. According to Wilks' theorem[44], as $H_0$ represents a special case of $H_1$, $\Delta\chi^2$ will follow a chi-squared distribution with two degrees of freedom in the large sample limit. Wilks' theorem thus provides the critical threshold value $\Delta\chi_{crit}^2$ corresponding to a desired CL, which allows us to quickly determine whether $H_0$ is compatible with $H_1$. This simplifies the sterile-neutrino analysis by providing a critical threshold $\Delta\chi_{crit}^2 = 5.99$, which corresponds to a 5% probability of obtaining such a deviation purely because of random fluctuations

under the null hypothesis. Thus, $\Delta\chi^2 > 5.99$ indicates a sterile-neutrino signal at 95% CL.

If Wilks' theorem were not applicable, the cumulative distribution function (CDF) of the $\Delta\chi^2$ test statistic would need to be computed for multiple combinations of the sterile-neutrino mass and mixing angle [$m_4^2$, $\sin^2(\theta_{ee})$] using Monte Carlo simulations, which would require several thousand times more computational effort. To ensure the applicability of Wilks' theorem to the analysis, the CDF of $\Delta\chi^2$ is numerically validated by performing calculations using Monte Carlo simulations for two sets of parameters. In the first case, [$m_4^2 = 0$, $\sin^2(\theta_{ee}) = 0$] corresponds to the null hypothesis, whereas in the second case [$m_4^2 = 55.66$ eV$^2$, $\sin^2(\theta_{ee}) = 0.013$] represents the best fit for the combined KNM1–KNM5 dataset. The CDFs and $\Delta\chi_{crit}^2$ values obtained through Monte Carlo simulations were compared with those predicted by Wilks' theorem.

For each of the sterile-neutrino mass and mixing-angle pairs, $\mathcal{O}(10^3)$ tritium β-decay spectra were generated by adding Poissonian fluctuations to the counts calculated using the model described in the section 'General model'. For each spectrum, the grid-scan method described in the main text is used to find the best-fit point and to compute $\Delta\chi^2$ using equation (6). For the case with $m_4^2 = 0$ eV$^2$ and $\sin^2(\theta_{ee}) = 0$, the numerically computed or empirical CDF closely follows the analytical CDF of the chi-squared distribution, as shown in Extended Data Fig. 9. Similarly, in the scenario with $m_4^2 = 55.66$ eV$^2$ and $\sin^2(\theta_{ee}) = 0.013$, the empirical CDF also aligns closely with the analytical CDF. In both cases, the observed threshold values at the 95% CL are consistent with the theoretical expectation of $\Delta\chi^2 = 5.99$. Specifically, for $m_4^2 = 0$ eV$^2$ and $\sin^2(\theta_{ee}) = 0$, the critical $\Delta\chi^2$ at 95% CL is $6.07 \pm 0.17$, and for $m_4^2 = 55.66$ eV$^2$ and $\sin^2(\theta_{ee}) = 0.013$, it is $6.07 \pm 0.20$. The uncertainties on the threshold values are calculated using the bootstrapping method. The combination of the CDF and the threshold values confirms the applicability of Wilks' theorem to the sterile-neutrino analysis described in the main text.

### Results compared with expected sensitivity

Figure 3 shows that the sensitivity contour, derived from simulations, intersects with the exclusion contour, obtained from the KNM1–KNM5 experimental data. For $\Delta m_{41}^2 < 30$ eV$^2$, the exclusion contour extends beyond the sensitivity contour, whereas at higher $\Delta m_{41}^2$ values, the exclusion contour oscillates around the sensitivity contour.

From the methodology outlined in the previous section and the description of $\mathcal{O}(10^3)$ fluctuated Asimov datasets generated under the null hypothesis in the context of Wilks' theorem, grid scans were performed to compute 95% CL contours. These scans distinguish between open contours (where $\Delta\chi^2 < 5.99$) and closed contours in which $\Delta\chi^2 \geq 5.99$ and the best-fit value would be distinguishable from the null hypothesis. The statistical band, encompassing the sensitivity contour obtained from Asimov simulations along with its $1\sigma$ and $2\sigma$ fluctuations, was subsequently reconstructed. As shown in Extended Data Fig. 10, the KNM1–KNM5 exclusion contour, derived from experimental data, falls within the 95% confidence region of the fluctuated simulated open contours.

This test confirms that the observed intersection between the sensitivity and exclusion contours is consistent with expected statistical fluctuations in the experimental data.

### Analysis with a free neutrino mass

In the main text, a hierarchical scenario is assumed with $m_4 \gg m_\nu$. Under this assumption, for the sterile-neutrino analysis, $m_\nu^2$ is set to 0 eV$^2$. However, for a small sterile-neutrino mass and large mixing, the active and sterile branches can become degenerate. A strong negative correlation between the active and sterile-neutrino mass for small $m_4^2 \leq 30$ eV$^2$ was reported in Sec. VI C of ref. 32. Hence, for small sterile-neutrino masses, it is interesting to perform the analysis considering the active-neutrino mass as another free parameter.

In Extended Data Fig. 11a, the 95% CL exclusion contours for the combined KNM1–KNM5 dataset are presented with statistical and systematic uncertainties incorporated. Two cases are analysed: the hierarchical scenario with the active-neutrino mass fixed to 0 $eV^2$ (scenario I, solid blue) and the scenario treating the squared active-neutrino mass $m_v^2$ as an unconstrained nuisance parameter (scenario II, dashed orange). The overlaid colour map indicates the best-fit values of $m_v^2$ across the parameter space in scenario II. For sterile-neutrino masses below 40 $eV^2$, partial degeneracy between the active- and sterile-neutrino branches weakens the exclusion bounds in scenario II relative to scenario I, with negative best-fit $m_v^2$ values highlighting the negative correlation. For larger sterile-neutrino masses, the exclusion contours of scenario II converge with those of scenario I as the correlation between the active- and sterile-neutrino masses decreases.

Apart from scenarios I and II, two additional scenarios were considered with specific restrictions on the squared active-neutrino mass. In scenario III, $m_v^2$ is treated as a free parameter but penalized by a $\pm 1$ $eV^2$ pull term centred around 0 $eV^2$. In scenario IV, $m_v^2$ is constrained by $0 \leq m_v^2 < m_4^2$, known as the technical constraint, first proposed by ref. 45. In Extended Data Fig. 11b, the 95% CL exclusion contours for the combined KNM1–KNM5 dataset are presented, incorporating only the statistical uncertainties.

The technical constraint in scenario IV (yellow dashed line) is of interest as its contour follows that of scenario I for lower squared sterile-neutrino masses (<40 $eV^2$) and that of scenario II for higher masses. Similar to Extended Data Fig. 11a, for only statistical uncertainties, the best-fit squared active-neutrino mass in scenario II is negative for sterile masses below 40 $eV^2$. As the squared active-neutrino mass is restricted to positive values in scenario IV, the resulting best-fit value is 0 $eV^2$, causing the contour of scenario IV to align with that of scenario I for low sterile-neutrino masses.

## KATRIN final sensitivity forecast

A projected final sensitivity for the KATRIN experiment at 95% CL is estimated using a net measurement time of 1,000 days, following the same prescriptions as in ref. 32. The background rate is expected to be 130 mcps for 117 active pixels, and the systematic uncertainties are based on the design configuration[2]. The primary update from the design configuration is the background rate, reflecting the current value. Moreover, the statistical variation is calculated using $\mathcal{O}(10^3)$ randomized tritium β-decay spectra with counts fluctuating according to a Poisson distribution. Sensitivity contours are calculated for each random dataset, and the $1\sigma$ and $2\sigma$ allowed regions that define the statistical fluctuations of the projected dataset are identified. A comparison with the exclusion contour from KATRIN for the first five measurement campaigns, see Extended Data Fig. 12, shows that for sterile masses $\Delta m_{41}^2 < 2$ $eV^2$ the final sensitivity contour is surpassed but not beyond the $1\sigma$ statistical uncertainty band. For $\Delta m_{41}^2 > 2$ $eV^2$, the sensitivity projection highlights the potential of KATRIN to probe the unexplored sterile-neutrino parameter space and to reach sensitivities of $\sin^2(2\theta_{ee}) < 0.01$. Given the observed difference between our current exclusion and the median sensitivity, future data may also yield less stringent limits due to statistical fluctuations. This is accounted for in our sensitivity projections, as shown by the $1\sigma$ and $2\sigma$ bands in Extended Data Fig. 12, which illustrate the expected range of statistical variation around the median.

The Taishan Antineutrino Observatory (TAO)[70], a satellite experiment of the Jiangmen Underground Neutrino Observatory (JUNO)[71], is primarily designed to precisely measure reactor antineutrino spectra but also sensitive to light sterile neutrinos, particularly at low values of $\Delta m_{41}^2$. The Precision Reactor Oscillation and Spectrum Experiment II (PROSPECT-II)[72], an upgraded version of the original PROSPECT detector[73], aims to enhance sensitivity to eV-scale sterile-neutrino oscillations. Together with the KATRIN experiment, these projects enable the exploration of sterile-neutrino mixing across a wide range of mass-splitting scales, allowing the probing of $\sin^2(2\theta_{ee}) \lesssim 0.03$ for $\Delta m_{41}^2 \lesssim 1{,}000$ $eV^2$, as shown in Extended Data Fig. 12.

## Data availability

The data and analysis inputs, as well as the results, are available at Zenodo[74] (https://doi.org/10.5281/zenodo.17205925).

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

**Acknowledgements** We acknowledge the support of Helmholtz Association (HGF), the Ministry for Education and Research BMBF (05A23PMA, 05A23PX2, 05A23VK2 and 05A23WO6), the doctoral school KSETA at KIT, Helmholtz Initiative and Networking Fund (grant agreement no. W2/W3-118), the Max Planck Research Group (MaxPlanck@TUM), Deutsche Forschungsgemeinschaft DFG (GRK 2149 and SFB-1258), the Excellence Strategy of Germany (EXC 2094-390783311), and the Graduate School Scholarship Programme of Deutscher Akademischer Austauschdienst (DAAD) in Germany; the Ministry of Education, Youth and Sport (CANAM-LM2015056) in the Czech Republic; the Istituto Nazionale di Fisica Nucleare (INFN) in Italy; the National Science, Research and Innovation Fund through the Program Management Unit for Human Resources and Institutional Development, Research and Innovation (grant no. B39G670017) in Thailand; and the Department of Energy through awards DE-FG02-97ER41020, DE-FG02-94ER40818,

DE-SC0004036, DE-FG02-97ER41033, DE-FG02-97ER41041, DE-SC0011091 and DE-SC0019304 and the Federal Prime Agreement DE-AC02-05CH11231 in the United States. This project has received funding from the European Research Council (ERC) under the European Union Horizon 2020 research and innovation programme (grant agreement no. 852845). We thank the computing support at the Institute for Astroparticle Physics at Karlsruhe Institute of Technology, Max Planck Computing and Data Facility (MPCDF), and the National Energy Research Scientific Computing Center (NERSC) at the Lawrence Berkeley National Laboratory.

**Author contributions** All authors contributed to this publication through various roles, including the design and construction of the experiment, performing theoretical calculations, software development, calibration of subsystems, operation, data acquisition and data analysis. The scientific results were discussed and approved by the full KATRIN Collaboration. The specific analysis presented in this Article, as well as the preparation of the manuscript, was carried out by a subgroup of authors (G. Gagliardi, C. Köhler, T. Lasserre, S. Mohanty and X. Stribl) and subsequently underwent an internal collaboration-wide review. All authors reviewed and approved the final version of the manuscript.

**Funding** Open access funding provided by Max Planck Society.

**Competing interests** The authors declare no competing interests.

**Additional information**
**Correspondence and requests for materials** should be addressed to T. Lasserre.

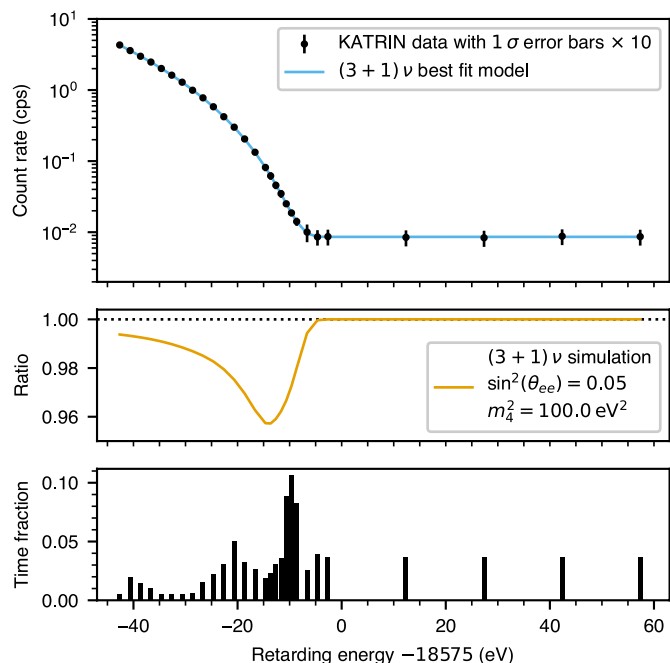

**Extended Data Fig. 1 | KATRIN data vs. model, simulated kink, and measurement-time distribution. a)** KATRIN data (KNM5, patch 0) with statistical uncertainties (error bars scaled by a factor of 10 for better visibility) and the best-fit model in the 3$\nu$+1 framework. The model includes contributions from both the active and sterile $\beta$-decay branches. **b)** Ratio of simulated spectra for the 3$\nu$+1 and 3$\nu$ frameworks, highlighting the kink-like signature of a sterile neutrino, most prominent near retarding energies $qU \approx E_0 - m_4$. **c)** Measurement-time distribution across retarding energies, showing the time fraction spent at each retarding energy during the integral spectrum acquisition.

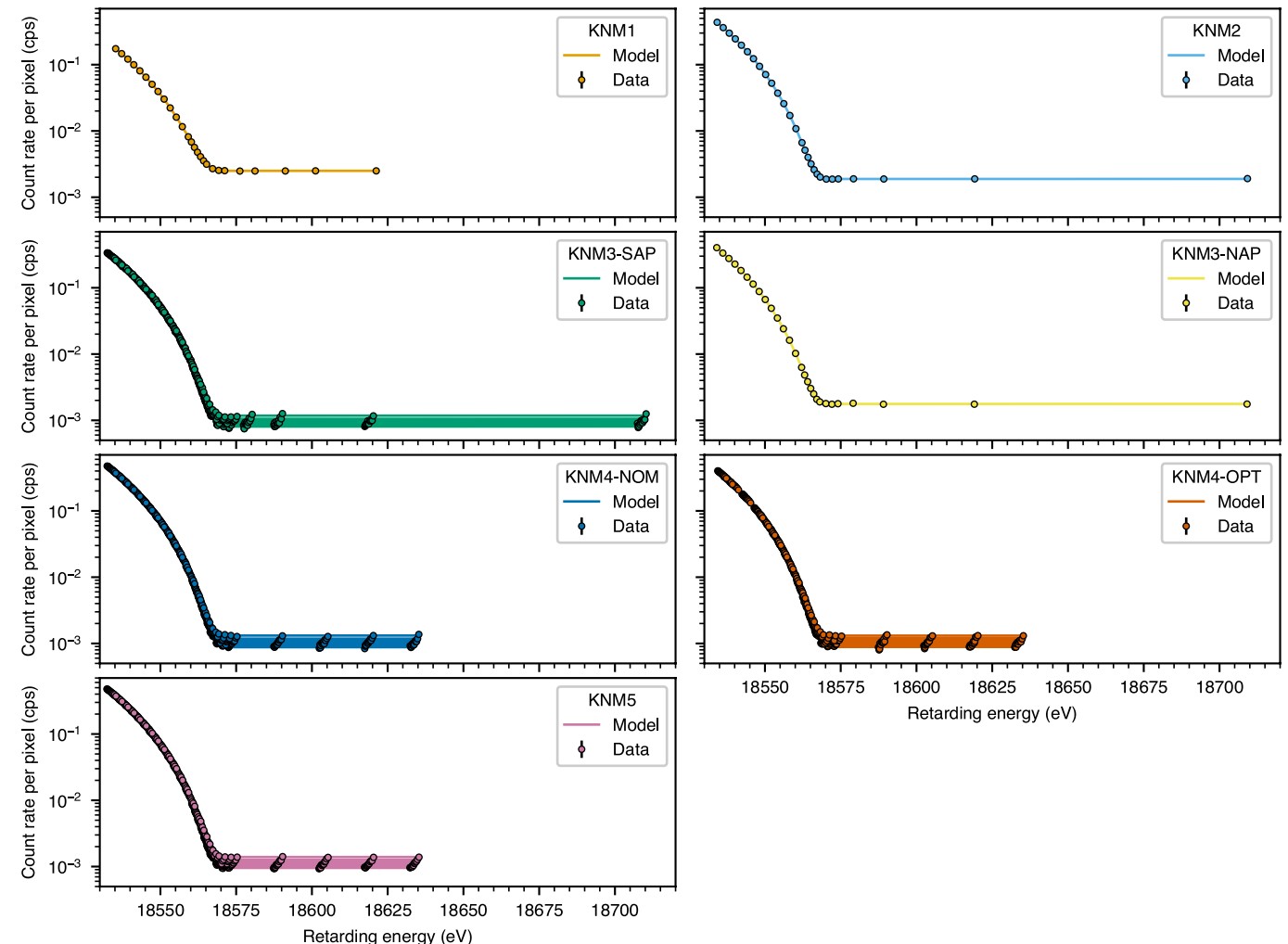

**Extended Data Fig. 2 | Measured spectra and global fit results across all campaigns.** The KNM3-SAP, KNM4-NOM, KNM4-OPT, and KNM5 datasets are divided into 14 detector patches. The squared mass of the sterile-neutrino state ($m_4^2$) and its mixing parameter ($\sin^2(\theta_{ee})$) are treated as common fit parameters across all campaigns.

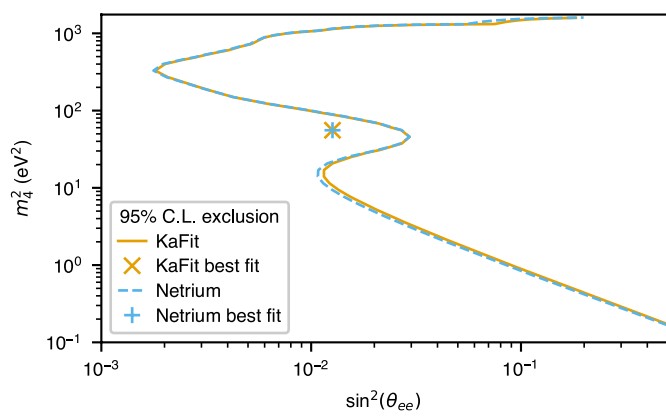

**Extended Data Fig. 3 | Comparison of exclusion curves from two independent analyses.** Comparison of 95% C.L. exclusion curves from the analysis of the first five KATRIN campaigns using two independent analysis procedures with $m_\nu^2 = 0\ eV^2$. Excellent agreement is observed regarding the excluded parameter space and the best-fit position.

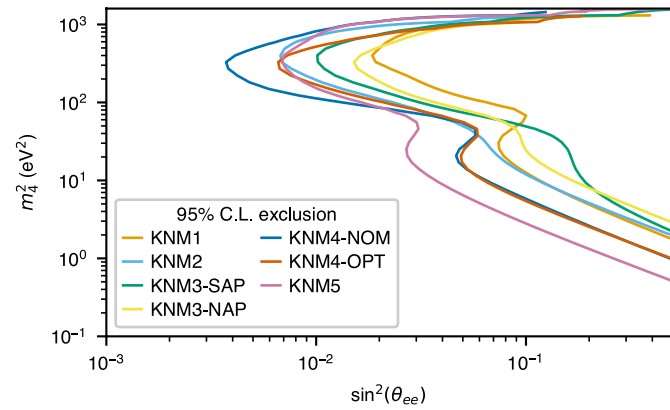

**Extended Data Fig. 4 | Sterile Neutrino Exclusion contours at 95% C.L. for individual KNM1–KNM5 campaigns.** The datasets include approximately 2.0 million electrons for KNM1, 4.3 million for KNM2, 2.5 million for KNM3, 10.2 million for KNM4, and 16.7 million for KNM5.

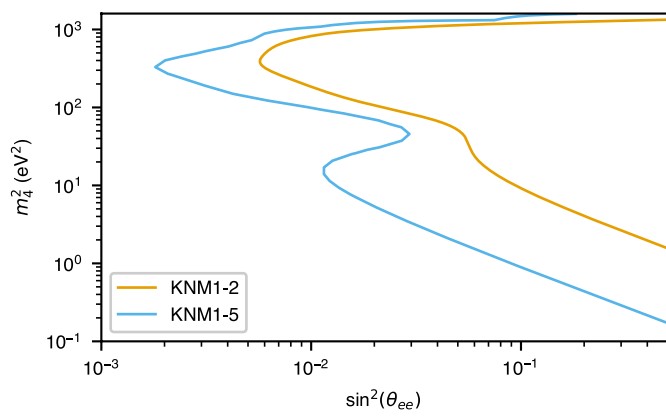

**Extended Data Fig. 5 | Comparison of exclusion contours from KNM1–5 vs. earlier KNM1–2 result.** Comparison of exclusion contours from the KNM1-5 analysis, based on 36 million electrons in the analysis interval, to the previous KNM1-2 result, which utilized 6 million electrons in the same analysis interval.

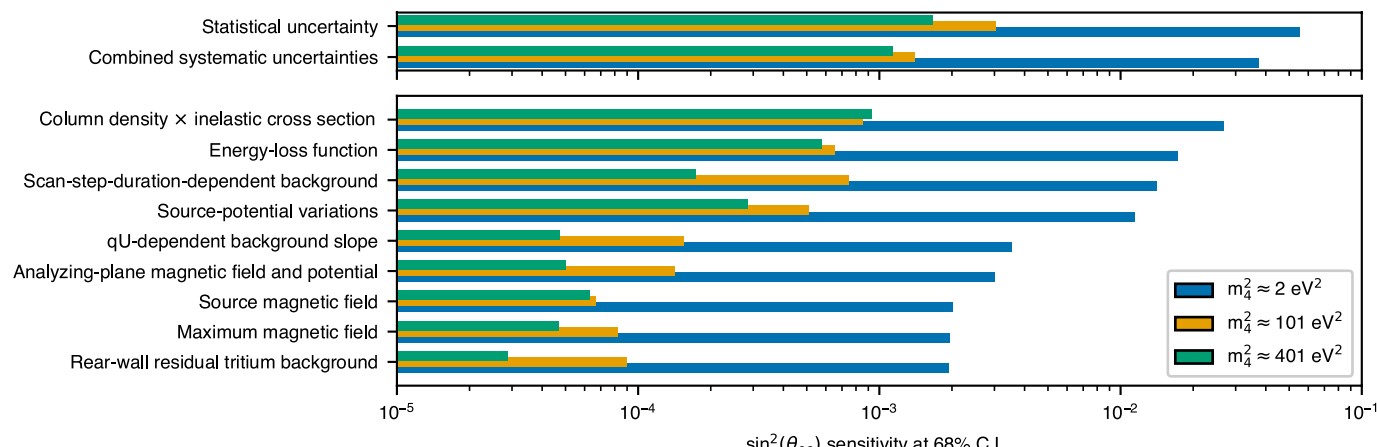

**Extended Data Fig. 6 | Contributions to the uncertainty on $\sin^2(\theta_{ee})$ from statistics and systematics at 68% C.L.** The figure comprises two panels: **Top)** Comparison of the combined systematic and statistical uncertainties for three representative values of $m_4^2$: 2 eV² (blue), 101 eV² (yellow), and 401 eV² (green). This illustrates that statistical uncertainty consistently dominates across all $m_4^2$ values. **Bottom)** Individual systematic effects are displayed, each represented by three coloured bars corresponding to the same $m_4^2$ values as in the top panel. The x-axis quantifies the uncertainty in $\sin^2(\theta_{ee})$ for each systematic effect, demonstrating their relative contributions to the total uncertainty.

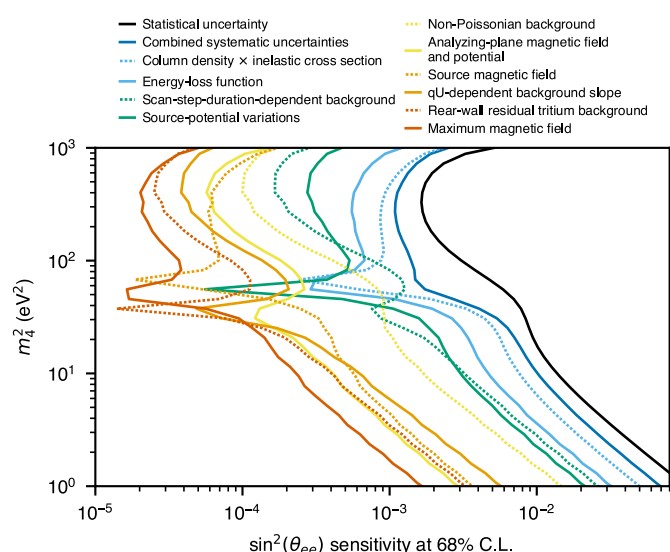

**Extended Data Fig. 7 | Breakdown of uncertainties.** Uncertainty breakdown for the combined dataset with $m_\nu^2 = 0$ eV$^2$. The plot focuses on the region $m_4^2 > 1$ eV$^2$. The 68.3% C.L. sensitivities on $\sin^2(\theta_{ee})$ for individual systematic effects and the statistical-only contour are displayed. Notably, all systematic effects are minor compared to the statistical uncertainty. The statistical uncertainty predominates over the combined systematic uncertainties across all $m_4^2$ values.

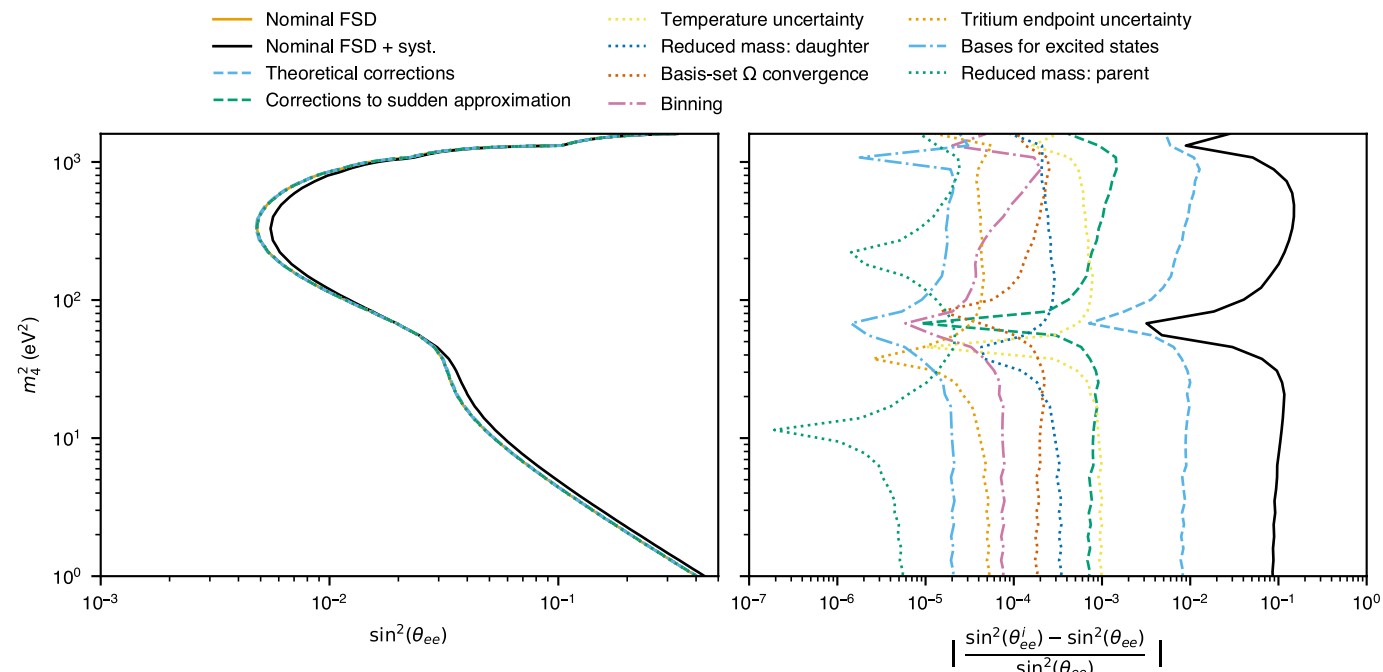

**Extended Data Fig. 8 | Impact of FSD systematics on the exclusion contour.**
**a)** 95% C.L. exclusion contours for datasets calculated under different FSD hypotheses. **b)** The absolute value of the normalized difference at each $m_4^2$ grid point between the 95% C.L. exclusion contours of the nominal FSD and the dedicated FSD datasets used in this analysis. The black solid line represents an Asimov dataset that includes all systematic uncertainties with nominal FSD, serving as a reference. Each contour illustrates the impact of a specific FSD systematic: dotted lines correspond to parameter uncertainties, dashed lines to theoretical approximations, and alternating dot-dashed lines to computational approximations.

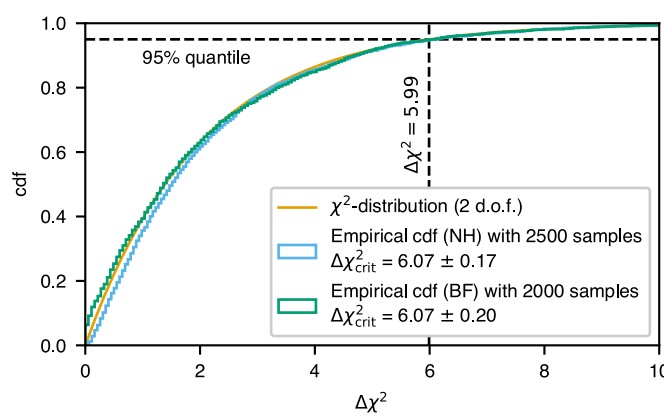

**Extended Data Fig. 9 | $\Delta\chi^2$ distributions from Monte Carlo compared to Wilks' theorem.** Cumulative Distribution Function (CDF) of $\Delta\chi^2$ for two sets of sterile-neutrino parameters compared to the analytical $\chi^2$ distribution with two degrees of freedom (orange line), as expected according to Wilks' theorem. The null hypothesis simulation (NH), $[m_4^2 = 0, \sin^2(\theta_{ee}) = 0]$, is shown in blue, while the best-fit simulation (BF), $[m_4^2 = 55.66 \, \text{eV}^2, \sin^2(\theta_{ee}) = 0.013]$, is shown in green. The corresponding empirical CDFs are calculated using Monte Carlo simulations.

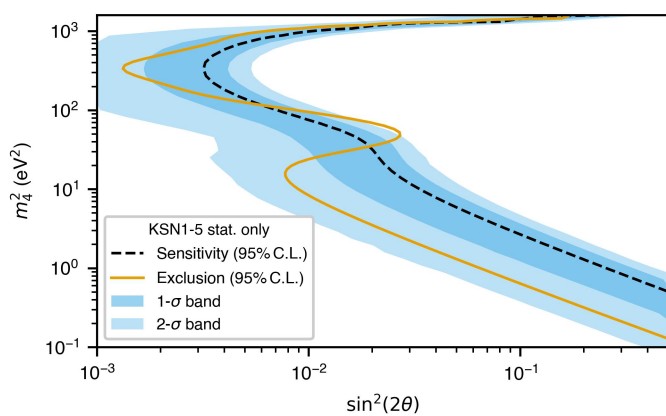

**Extended Data Fig. 10 | KNM1–5 exclusion vs. sensitivity band, showing consistency with expected fluctuations.** Comparison of the KNM1-5 exclusion contour (data) with the 95% C.L. statistical band from simulated sensitivity contours, showing consistency with the statistical fluctuations.

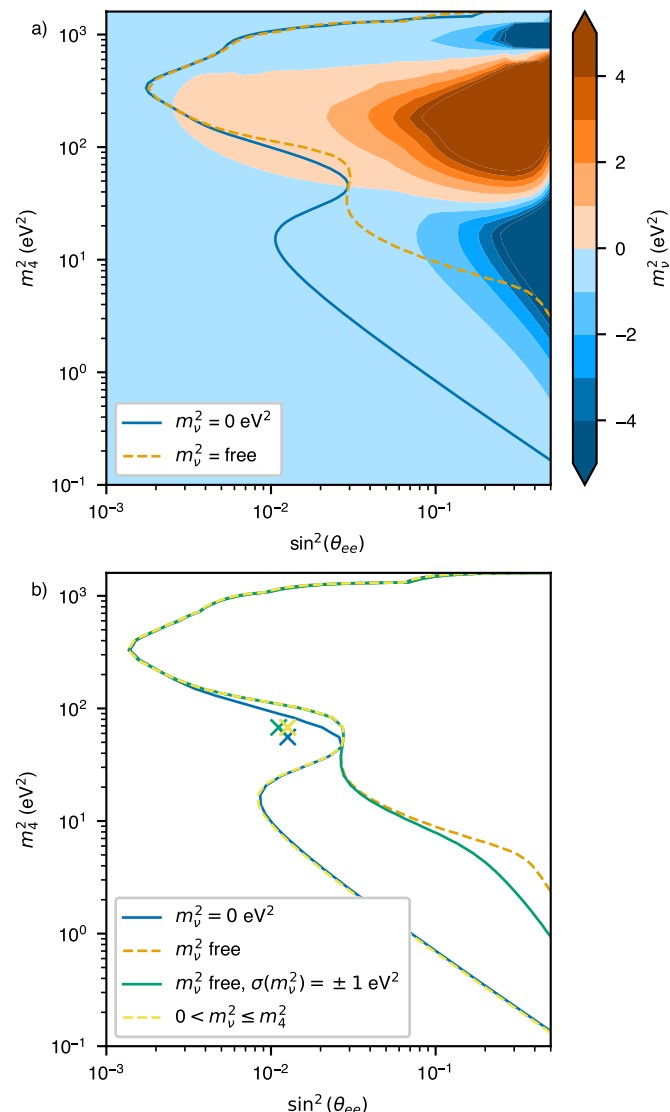

**Extended Data Fig. 11 | Effect of different treatments of $m_\nu$ on the exclusion contour. a)** 95% C.L. exclusion contour of KNM1-5 including statistical and systematic uncertainties with active-neutrino mass fixed to 0 eV$^2$ and left unconstrained in the analysis. The colour map describes the best-fit active-neutrino mass values. **b)** 95% exclusion contour of KNM1-5 including statistical uncertainties and different scenarios for the treatment of the active-neutrino mass.

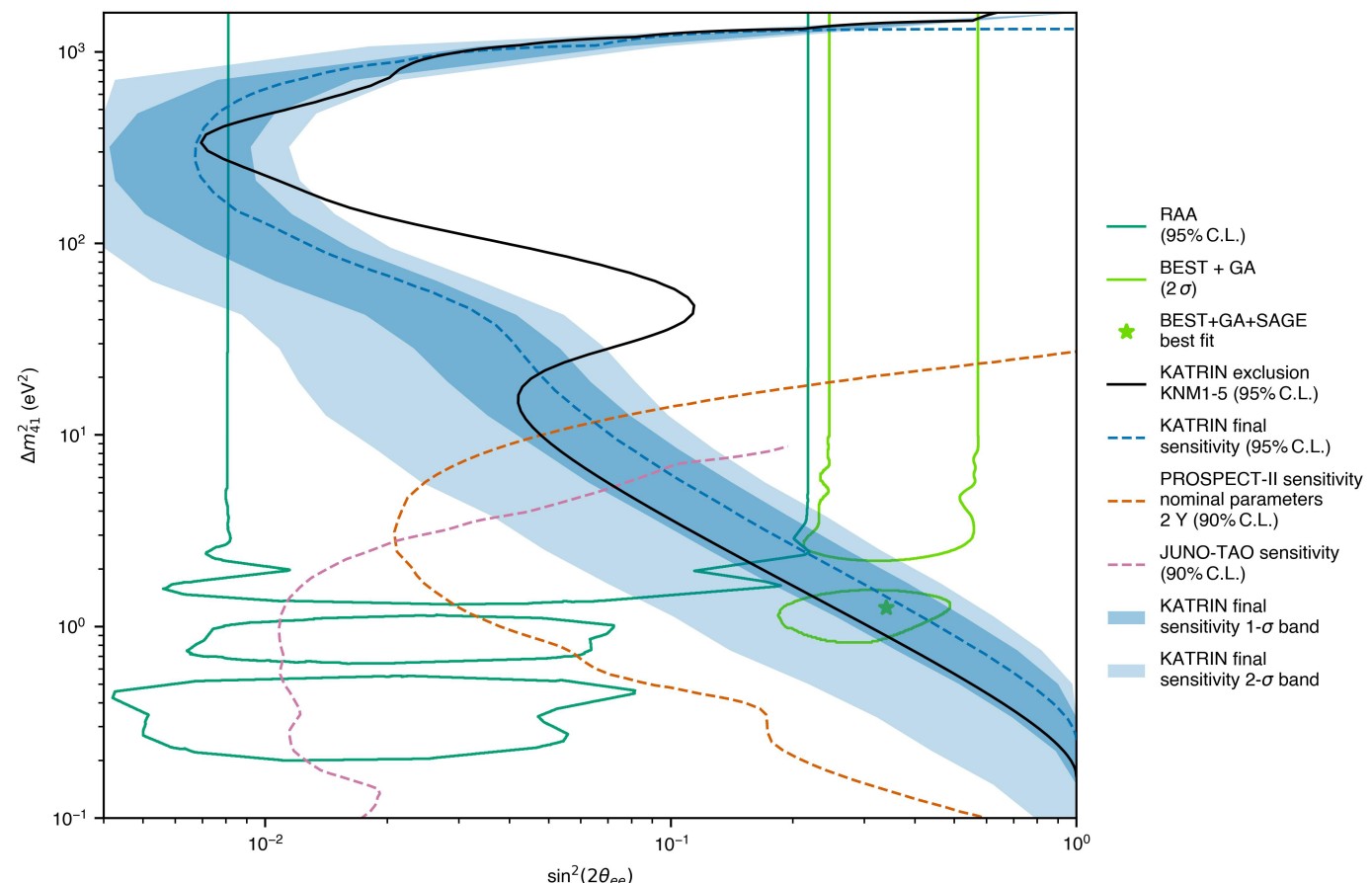

**Extended Data Fig. 12 | Projected KATRIN sensitivity with the full 1000-day dataset.** The dashed blue line shows the 95% C.L. sensitivity contour, while the dark and light blue bands indicate the 1- and 2-$\sigma$ statistical uncertainty. The black line represents the KNM1-5 exclusion result (this work). The figure highlights significant potential improvements in the sterile-neutrino search with the final dataset expected after 2025.

**Extended Data Table 1 | Summary of the main characteristics of the KATRIN measurement campaigns**

| KNM Campaign | Year/Month | Days | Analysing Plane | $T_{\text{WGTS}}$ (K) | Bkg. (cps) | $\rho\,d$ $(\times 10^{21}\ \text{m}^{-2})$ |
|---|---|---|---|---|---|---|
| 1 | 2019/04-05 | 35 | nominal | 30 | 0.29 | $1.08 \pm 0.01$ |
| 2 | 2019/10-12 | 45 | nominal | 30 | 0.22 | $4.20 \pm 0.04$ |
| 3-SAP | 2020/06-07 | 14 | shifted | 79 | 0.12 | $2.05 \pm 0.02$ |
| 3-NAP | 2020/07 | 14 | nominal | 79 | 0.22 | $3.70 \pm 0.04$ |
| 4-NOM | 2020/09-10 | 49 | shifted | 79 | 0.13 | $3.76 \pm 0.04$ |
| 4-OPT | 2020/10-12 | 30 | shifted | 79 | 0.13 | $3.76 \pm 0.04$ |
| 5 | 2021/04-06 | 72 | shifted | 79 | 0.14 | $3.77 \pm 0.03$ |

$\rho d$ and $T_{\text{WGTS}}$ denote the column density and temperature in the gaseous tritium source, respectively, with precision within the significant digits. Bkg. represents the background rate in count per second (cps).