## [Peer Review File · Nature]

Sterile-neutrino search based on 259 days of KATRIN data

Corresponding Author: Dr Thierry Lasserre

Version 0:

Reviewer comments:

Referee #1

(Remarks to the Author)
Key results.

This work contains extremely important results on the search for a fourth species of massive neutrinos, the so called sterile neutrinos. Even a null result like the one presented here is extremely useful to shed light on the debate about the existence of a sterile neutrino. A number of experiments of totally different types are in fact showing inconsistent results with the standard neutrinos model with the three weak interacting families

Validity.

I believe that the details provided by the authors are enough to judge the experiment validity that is in fact high. The Method parts are extensively describing a lot of details with a good reference apparatus.

However - as shown in Fig.9 - the level of combined systematics is almost at the level of the statistical uncertainty. Being this an integral measurement of the beta spectrum - that is fixing a voltage threshold and counting all events above that threshold - it would be interesting to have more details on how this can spoil the detection of a kink - that is the signature of the massive sterile neutrino.

If the voltage stability is known at some level, this uncertainty can lead to a smearing of the kink. Therefore there could be some inefficiency in detecting the kink and this would shrink the exclusion region. Can the authors address this with some Asimov experiment test? or did they do this and can they explain with some more details ?

A source of systematics might be the stability of the p-i-n diode. Is the diode efficiency measured independently so not to affect the spectrum reconstruction ? I cannot judge whether this is addressed in section 8.

Are in general the systematics independent of the sub-dataset ? The authors are eventually stacking together all the results of the sub-datasets. But there might be one or more with a larger systematics error that might lead to less significant exclusion (see KNM4 and the problem described at line 890). Did they investigate this ?

Also (from line 580) background is described. It is not clear to me how this can be energy dependent. If this were the case it might distort the results. How do the authors control this ?

Do they have a graph of the spectrum of background events?

Originality and significance.

The results are of great interest for the neutrino physics community as described in Fig.3 where the null result of this experiment is compared with the positive results of others.

Data & methodology.

Supplementary information provides a good amount of information about the quality of the data analysis.

Appropriate use of statistics and treatment of uncertainties (if applicable).

The authors are employing a sound statistical method already explained in other publications

Conclusions. Are the conclusions and data interpretation robust, valid, appropriate and reliable?

Yes they are

Suggested improvements.

Please list additional experiments or data that could help strengthen the work in a revision.

I have very few:

line 193 : $\sin^2(\theta_{ee})$ is used while at line 214 $\sin^2(2\theta_{ee})$ is given. For the casual reader this can be confusing. Maybe a word of explanation would be valuable

line 201: a "few meter" baseline is mentioned and then line 202 a "lower" baseline to 10 to 100 meters is quoted. I find this confusing.

References. Does the manuscript reference previous literature appropriately?

Yes it does

Clarity and context. Is the abstract clear and accessible? Are the abstract and introduction appropriate?

yes they are

(Remarks on code availability)

Referee #2

(Remarks to the Author)

The key result of this work is an improved limit on sterile neutrino parameter space from beta decay spectral data. The data and experiment are unique and therefore the result is an important contribution to the field. This result will interest the broader particle and nuclear physics communities. The authors gave a thorough discussion on these exciting and original results and provided an overview of the various methods that went into calculating the final sterile neutrino exclusion. This manuscript provides a good starting point for how the final sterile neutrino exclusion sensitivity was calculated, leaving more detailed explanations of many of these subsections for future follow up papers. The paper is well written and structured and should be published, but with some edits or additions for clarity.

Some data was excluded due to quality issues (near lines 358 and 385). An additional sentence or two explaining why the quality led to exclusion is warranted.

In line 215, the word contentious is moderately inflammatory. Replace with the word unconfirmed.

Line 238: the use of the word "experiment's" is confusing within the context of the rest of the paragraph. Are the authors referring to the systematic uncertainties of a lone experiment? Or did they mean experiments'? If it's only one experiment, be clear about which one.

Within the paragraph that begins at Line 390, the fit description says that the 3 active squared neutrino mass is fit (line 392), but later it says that parameter is set to zero (line 397). Some editing to make it clear what was done would be useful.

Line 530: The data presented was collected prior to 2021. Providing a sentence regarding what data has been taken since then would be useful in a broader context, especially given the projection section 13.

The star symbol represented in Fig. 6 should be explained in the caption?

For the fits, the resulting value of E_0 would be interesting to include and how fixing it to the Penning trap result would, or would not, affect the sterile parameter constraints. Does the uncertainty in E_0 contribute to the systematic uncertainty? Adding the best fit values of E_0 , m_{ν} , to Table 1 would be nice to see.

The middle panel of Fig. 4 is very useful --more useful than the panels in Fig. 5. Are there uncertainties on the points in that panel? If so, describe them. If not, consider adding them.

The statistical limit is significantly better than the quoted sensitivity. That implies a statistical fluctuation. In Fig. 4 middle, evidence for a sterile neutrino is a dip in the ratio. For these data, it seems a fluctuation led to a bump which provides the improvement over the sensitivity. Perhaps include those plots to give one a feel for any potential fluctuations.

The curves representing the systematic uncertainties within the excluded parameter space depicted in Fig. 10 have extended features near 30-50 eV². Can the authors explain the origin of those features?

The qualitative conclusions (“excludes much of the parameter space”, “rules out ... with high confidence”) to be a bit strong. The sensitivity is to the right of a large fraction of Neutrino-4 and the key Gallium anomaly space. That is, although the limit excludes much of the sterile neutrino parameter space indicated by previous experiments, the sensitivity does not. With the significant difference between the limit and sensitivity, it is possible that the limit will worsen with further data. The authors should add some discussion to provide a caveat regarding the sensitivity-limit difference and the comparison to other experimental claims.

There are some citations that should be included or updated. These include:

The SAGE reference [11] is about the solar neutrino results and not the Cr-51/Ar-37 data. Better references are: 51Cr: PRC 59 (1999) 2246, 37Ar: PRC 73 (2006) 045805.

A Recent review article on the gallium anomaly should be cited. Prog. Part. Nucl. Phys 134 (2024) 104082.

Ref. 25 is incomplete.

The capitalization of KATRIN is not uniform within the references. (see for example, 23, 42, 44)

A reference to the concept of an Asimov data set should be provided.

The IceCube, MINOS, SuperK, T2K and NOvA collaborations have recently published papers exploring light (around 1 eV² and below) sterile neutrinos. Perhaps these don't fit into the introduction to this work closely enough due to the different mixing channel or because they are null results, but the authors might consider a sentence and appropriate citations.

(Remarks on code availability)

The editor mentioned there was going to be a codebase/dataset made available to the community as part of this paper and asked for an assessment of usability by the reader. However, the Zenodo link that was sent as follow-up was specifically for the previous mainline KATRIN result for the direct neutrino mass measurement and not this specific sterile neutrino search (<https://zenodo.org/records/13644900>). This analysis is complex and requires large scale computing resources to calculate the various final exclusion curves therefore data sharing is complicated. Even so, it would help the community to release some part of the final high-level analysis/sensitivity (final spectra and exclusion curves, e.g.) as an open-source dataset with instructions included with the raw data. The Zenodo link only contains the raw json files and requires that the user go to the last section of the mainline KATRIN paper to find an explanation.

The open source data releases of the CDMS WIMP dark matter search experiment could serve as an example: <https://supercdms.slac.stanford.edu/science-results/data-releases>. While CDMS also has large scale data sets with multiple cuts, most of these data releases allow others to download and reproduce most of the final data curves that go into calculating the final sensitivity limit. At the same time something that is vital for these data releases is the accompanying documentation/explanations again see CDMS data releases that allow for other members of the community to compare their calculations with the potential sensitivity limits.

Referee #3

(Remarks to the Author)

I co-reviewed this manuscript with one of the reviewers who provided the listed reports.

(Remarks on code availability)

Referee #4

(Remarks to the Author)

A - Summary of key results. KATRIN has conducted a search for sterile neutrinos in KNM 1-5 data, taken between 2019 and 2021, increasing statistics over their previous search by a factor of 6. Importantly, the limits are unmatched at large mass-squared values and completely rule out the Neutrino-4 best fit point (already largely ruled out by PROSPECT), a large region the Gallium Anomaly/BEST parameter space, and large regions of the Reactor Antineutrino Anomaly parameter space. Results are complementary to reactor antineutrino experiments, which generally probe lower mass-squared regions. This is a high quality and impactful result worthy of publication.

B - Originality and significance. KATRIN is the only experiment currently capable of conducting this search at the achieved sensitivity for large mass-squared. The results are a significant improvement over their previously published results [Phys. Rev. D 105, 072004 (2022)] and address important regions of parameter space.

C - Data & Methodology. The search for a spectral distortion in the beta spectrum is a well-known method to search for sterile neutrino mixing. Data quality has been thoroughly vetted and also serves the neutrino mass analysis. The KATRIN

collaboration has clearly done a lot of work to understand and mitigate sources of background and characterize detector response over their 5 KNM campaigns, and this is well described in Section 3.

D - Statistics. Treatment of statistics seems appropriate. Uncertainties are statistics dominated over the full range of mass-squared values tested. Wilks' theorem is applied, and its applicability is justified through comparison with a Monte Carlo-based cross check at the best fit point, as detailed in Section 10.

E - Conclusions. Use of blinding and the validation of the analysis using two independent methods add confidence. The collaboration's identification and remediation of faulty data stacking during KNM 4 is well described. Results appear robust and lay a strong foundation for a future search with the full KATRIN statistics, not to mention the planned TRISTAN upgrade to the KATRIN apparatus.

F - Suggested improvements. No major corrections needed, but two small suggestions:

a. I question the need for Figure 2 as an illustration in the main text – it doesn't seem to add much to the discussion, and the method for drawing contours is standard in the field.

b. The color choices in Figure 3 make the different contours (particularly for DANSS, STEREO, and PROSPECT) difficult to interpret. In the legend, GA is usually interpreted to include both GALLEX and SAGE, so BEST+GALLEX+SAGE best fit is somewhat confusing. STEREO and PROSPECT should be in all caps in the legend of the figure, and I recommend including explicit references for each of the curves in the figure that do not come from this work.

G - Referencing. The null results of sterile neutrino searches with IceCube should also be referenced, for example [Phys. Rev. Lett. 133, 201804 (2024)].

H - Clarity and context. The introduction could be written more cohesively to clearly articulate where KATRIN's search fits into the landscape of sterile neutrino searches. In many ways, the radioactive source anomalies are the best motivation for KATRIN's search, and they provide the most direct comparison. The RAA is complicated, and while it is mentioned that the Neutrino-4 result is controversial, the null result of the PROSPECT experiment isn't mentioned until 3 paragraphs later. It might be best to discuss each anomaly fully, including null results, in turn in order to give a sufficiently nuanced view of the current state of the field. IceCube results should be included in the discussion of LSND and MiniBooNE.

(Remarks on code availability)

Referee #5

(Remarks to the Author)

The KATRIN collaboration is presenting its most recent limit, based on 259 days of data, on a sterile neutrino in the 1-1000eV mass range. Sterile neutrinos in this mass range are motivated both from broader theoretical considerations as well as from experimental indications. The KATRIN bound likely will remain the most stringent and most model-independent bound in that mass range for quite some time, only to be superseded by its own final limit. Thus the paper is a good fit for Nature.

The paper is well written and the methods section is very detailed. The statistical treatment is sophisticated and it is nice to see that they made the effort to actually test the applicability of Wilks' theorem and find that indeed it holds.

I think that assuming the active neutrino mass to be zero for extracting a limit on the sterile neutrino mass is not logically consistent. KATRIN is an experiment to measure neutrino masses and arguably the most sensitive one, therefore, all information on neutrino masses used in the fit should come from its own data. This is for instance the approach taken in the KATRIN main result paper on the active neutrino mass, where negative mass squared values are allowed in the fit. Indeed, KATRIN is the first tritium experiment to find mass squared values that are consistent with being zero, see Fig 3 of <https://doi.org/10.1126/science.adq9592>. Therefore, the dashed yellow line ($m\nu^2$ free) of Fig 15 in the current manuscript should be the main limit shown in Fig 3. This is important because KATRIN is the most direct, i.e. least model-dependent, test of the sterile neutrino hypothesis. Adding the constraint on the active neutrino mass squared makes this result less valuable.

In terms of framing this result as addressing the so called reactor antineutrino anomaly (RAA), I would like to stress that there is an emerging consensus among experts that the RAA is not due to a sterile neutrino but due to a normalization problem with the beta spectrum data collected in the 1980/90s at the Institut Laue-Langevin. The introduction contains some hedging phrases acknowledging this, but the figures prominently display the RAA allowed region. So I would like to suggest removing the RAA region from Figure 3.

The data release on <https://doi.org/10.5281/zenodo.13644899> is somewhat minimal and it's unclear if anyone would be able to reproduce Fig 3 from that data. I would like to suggest in addition to also release a χ^2 table spanning the range of parameters in Fig 3. That at least would ensure proper inclusion in global fits. Further, maybe it is conceivable to augment the data release with a simple χ^2 -model including the leading systematics (maybe in the form of a covariance matrix)? This would allow the wider pheno community to test a range of new physics scenarios.

(Remarks on code availability)

see above

Version 1:

Reviewer comments:

Referee #1

(Remarks to the Author)

The authors replied to all my queries with a satisfactory number of details. I went through them and to the various modifications they are proposing to their draft and I believe now this version can be published.

(Remarks on code availability)

Referee #2

(Remarks to the Author)

We are satisfied with the responses of the collaboration to our concerns and questions. We thank the KATRIN collaboration for their efforts. We recommend publication. We do have two minor issues regarding the data release, which we include within that review entry.

(Remarks on code availability)

We appreciate the complexity of doing a full software release and understand the need to keep the codebase internal for now. There were only a few follow up issues related to the zenodo document to address: 1) The provided link for <https://zenodo.org/records/15860995> was actually initially directed towards a dead zenodo repository, the text has the correct link, but the actual hyperlink was directed towards <https://zenodo.org/records/xxxxxxx>, please make sure to update the correct link in the paper. We also suggest to include a short README/pdf file within the zenodo repository explaining what all the json files contain as well as any extended commentary on those data sets that could be of use to the pheno community when they superimpose your results onto their latest works.

Referee #3

(Remarks to the Author)

I co-reviewed this manuscript with one of the reviewers who provided the listed reports.

(Remarks on code availability)

I co-reviewed this manuscript with one of the reviewers who provided the listed reports. Comments on code availability provided on joint report.

Referee #4

(Remarks to the Author)

The KATRIN sterile neutrino search presents a high quality and impactful result worthy of publication in Nature. Comments have been addressed satisfactorily. In particular, I appreciate the collaboration's substantial reworking of the introduction, which is very clear.

(Remarks on code availability)

Referee #5

(Remarks to the Author)

From the collaboration's rebuttal:

"We believe that presenting the results with $mv^2 = 0$ as our main exclusion contour remains the most appropriate choice for this sterile neutrino analysis, for the following reasons:

If mv is left free, then at each point on the sterile parameter grid ($m_4^2, \sin^2\theta_{ee}$) the fit can yield a different value for mv , and therefore for the masses m_1, m_2, m_3 . This means we cannot consistently construct Δm_{41}^2 to compare our results with neutrino oscillation experiments. Fixing $mv^2 = 0$ ensures a clear and consistent mapping in the standard oscillation parameter space. We have also shown (in Methods Section 12 and Fig. 15) that applying a minimal technical constraint, $0 < mv^2 \leq m_4^2$, leads to results that are essentially the same as fixing $mv^2 = 0$."

When asking the data to constrain a sterile neutrino one is asking to constrain 2 masses instead of one and thus, the result overall is weaker — more parameters = weaker constraint. I accept the explanation that the technical issues arising from the two pseudo-branches switching their position and hence imposing $mv^2 \leq m_4^2$ make sense, and I would do the same. It is legitimate to do so because no part of parameter space is precluded by this choice in the fit; it is indeed a mere "technical" constraint. However, requiring that both mass squared values are positive is constraining the parameter space and this

amounts to imposing a physics prior on the analysis. And as results in Fig 15 show, this prior leads to a much tighter constraint, so it actually does matter. In a Bayesian spirit, I would say, if the choice of the prior matters so much then maybe the data by itself does not allow for such a strong inference. If the collaboration had decided for their main neutrino mass analysis to impose a prior of positive mass squared values, the argument for the present paper would be more straightforward and I would tend to agree with the authors for this paper (but not the main analysis), but of course this is not the case. The reason, I believe, why the collaboration did not impose this positivity prior for the main analysis is that there are many systematic errors that can push the mass squared to negative values and historically that has been found to be true for all previous tritium experiments. In my view, the fact that the main analysis finds a slightly negative best fit within one standard deviation from zero is demonstrating that KATRIN has good control of its systematics. Now, constraining the squared masses to be positive results in an unwarranted reduction of systematic uncertainty. Therefore, I stand by my earlier request to not use the positivity constraint for the main result. All other aspects raised by me and the other referees have been addressed and I thank the authors for their efforts.

(Remarks on code availability)

Answers to “KSN1-5 referee comments from Nature (First Round)”

Referee 1

- **Question/comment:** *However - as shown in Fig.9 - the level of combined systematics is almost at the level of the statistical uncertainty. Being this an integral measurement of the beta spectrum - that is fixing a voltage threshold and counting all events above that threshold - it would be interesting to have more details on how this can spoil the detection of a kink - that is the signature of the massive sterile neutrino. If the voltage stability is known at some level, this uncertainty can lead to a smearing of the kink. Therefore there could be some inefficiency in detecting the kink and this would shrink the exclusion region. Can the authors address this with some Asimov experiment test? or did they do this and can they explain with some more details ?*

Answer: The KATRIN high-voltage system is extremely stable. As shown in our dedicated study (J. High Volt. 1, 234 (2022)), the fluctuations are smaller than 2 ppm, that's below 40 mV at 18.6 kV. The sterile neutrino signal is not just a sharp kink: it also causes a global distortion of the spectrum below the kink. So the sensitivity is not lost even if the kink is slightly smeared. For instance, a sterile kink for a 10 eV neutrino affects a broad energy window (>20 eV), so the possible smearing is negligible. In addition, we explicitly included source potential variations (similar to HV potential variation) in our previous sterile neutrino analysis (PRD 105, 072004 (2022), arXiv:2201.11593) and found them to have only a minor effect.

- **Question/comment:** *A source of systematics might be the stability of the p-i-n diode. Is the diode efficiency measured independently so not to affect the spectrum reconstruction ? I cannot judge whether this is addressed in section 8.*

Answer: The detector response is regularly checked using calibration runs with an external ^{241}Am source to ensure stability over time. The overall detection efficiency is about 95%, with only a few percent uncertainty and similar variation between pixels. These values are stable and effectively constant across the energy region of interest. In addition, any efficiency effects that do not depend on the retarding potential do not impact the sterile neutrino analysis, since each spectrum is fitted with a free normalization and all detector pixels are treated in a combined fit (see PRD 104, 012005, 2021).

Statement for publication:

\cor{This detector is regularly calibrated with an ^{241}Am source to ensure stable performance. Its efficiency is about 95%, with only small variations between pixels that remain constant over time. Effects that do not depend on the retarding potential are accounted for by the free normalization combined fit across group or patches of pixels~\cite{Aker2021_PRD104_012005}.}

```
@article{Aker2021_PRD104_012005,
  author    = {M. Aker and K. Altenmueller and C. AMSLER and et al.},
  title     = {Analysis methods for the first KATRIN neutrino-mass
measurement},
  journal   = {Phys. Rev. D},
  volume    = {104},
  number    = {1},
  pages     = {012005},
  year      = {2021},
  month     = jul,
  doi       = {10.1103/PhysRevD.104.012005},
}
```

- **Question/comment:** *Are in general the systematics independent of the sub-dataset ? The authors are eventually stacking together all the results of the sub-datasets. But there might be one or more with a larger systematics error that might lead to less significant exclusion (see KNM4 and the problem described at line 890). Did they investigate this ?*

Answer: The full dataset was divided into five measurement campaigns to account for possible changes in experimental conditions over time. Each campaign represents a period with stable systematics and detector performance. We checked the stability of key input parameters used for systematic modeling through dedicated calibration and monitoring measurements to confirm that combining sub-datasets within a campaign is justified. The issue identified in the KNM4 campaign (see line 890) was specific to that dataset and does not affect the others. It was eventually properly included in the modeling of KNM4. Finally, the quality of the neutrino mass and sterile neutrino fits, including the pull terms, show no sign of missing or unaccounted systematic effects.

- **Question/comment:** *Also (from line 580) background is described. It is not clear to me how this can be energy dependent. If this were the case it might distort the results. How do the authors control this ?*

Answer: The transmission and detection probabilities of background electrons may slightly depend on the retarding-potential setting, potentially causing a retarding-potential dependent background. We account for this by including an additional slope component. Through dedicated calibration measurements, where the background is measured at different retarding potential levels, we constrain the slope parameter, assuming a linear relationship between background and retarding potential. The measured slopes are (0.9 ± 3.2) mcps/keV in the NAP setting and (1.1 ± 0.7) mcps/keV in the SAP configuration. This effect is included in our analysis as a systematic uncertainty (see “qU-dependent background slope” in Fig. 10 for instance). We treat the background slope as a nuisance parameter constrained by the values mentioned above. The shape of the background slope is illustrated in Figure 12 in the supplementary material section of “Direct neutrino-mass measurement based on 259 days of KATRIN data”

Statement for publication (in the Methods Section 2):

\cor{Additionally, the transmission and detection probabilities of background electrons may slightly depend on the retarding-potential setting, potentially causing a retarding-potential dependent background. This effect is addressed in the analysis through an additional slope component, constrained by dedicated background measurements~\cite{Katrin:2024tvj}.}

```
@article{Aker2025_KATRIN259days,
  author    = {M.~Aker and D.~Batzler and A.~Beglarian and J.~Behrens
and J.~Beisenkötter and M.~Biassoni and B.~Bieringer and Y.~Biondi and
F.~Block and S.~Bobien and M.~Böttcher and B.~Bornschein and
L.~Bornschein and T.~S.~Caldwell and ... (KATRIN Collaboration)},
  title     = {Direct neutrino-mass measurement based on 259 days of
KATRIN data},
  journal   = {Science},
  volume    = {388},
  number    = {6743},
  pages     = {180–185},
  year      = {2025},
  month     = apr,
  doi       = {10.1126/science.adq9592},
  eprint    = {2406.13516},
  archivePrefix = {arXiv},
  primaryClass = {nucl-ex}
}
```

- **Question/comment:** *Do they have a graph of the spectrum of background events?*

Answer: The shape of the background including the retarding potential slope (“qU slope”) and the scan-step-duration-dependent background (“Penning increase”) is illustrated Figure 12 in the supplementary material section of “Direct neutrino-mass measurement based on 259 days of KATRIN data”. Note the zoom-in on the y-axis due to the small magnitude of the non-constant background effects.

```
@article{Aker2025_KATRIN259days,
  author = {M.~Aker and D.~Batzler and A.~Beglarian and J.~Behrens
and J.~Beisenkötter and M.~Biassoni and B.~Bieringer and Y.~Biondi and
F.~Block and S.~Bobien and M.~Böttcher and B.~Bornschein and
L.~Bornschein and T.~S.~Caldwell and ... (KATRIN Collaboration)},
  title = {Direct neutrino-mass measurement based on 259 days of
KATRIN data},
  journal = {Science},
  volume = {388},
  number = {6743},
  pages = {180–185},
  year = {2025},
  month = apr,
  doi = {10.1126/science.adq9592},
  eprint = {2406.13516},
  archivePrefix = {arXiv},
  primaryClass = {nucl-ex}
}
```

- **Question/comment:** *Please list additional experiments or data that could help strengthen the work in a revision.*

Answer: Since the LSND result in 1994 and especially after the combined reactor and gallium anomaly publication in 2011, there have been many important searches for light sterile neutrinos. In our work, we decided to highlight experiments that are primarily sensitive to the channel involving $|\text{Ue4}|^2$ to keep the discussion focused and easier to follow for readers who are not experts in sterile neutrino physics. We now modified the text and added the following paragraph with additional references to the main article:

“

Other significant anomalies include observations from the LSND experiment~\cite{LSND:2001aii}, which ran during the 1990s, and detected an excess of electron antineutrino events in a muon antineutrino beam. This finding was interpreted as potential evidence for sterile-neutrino involvement.

`\cor{Around the same time, the KARMEN
 experiment~\cite{KARMEN:2002zcm}, which probed the same oscillation
 channel, reported no such excess.} Later, the MiniBooNE
 experiment~\cite{MiniBooNE:2007uho,MiniBooNE:2008yuf,MiniBooNE:2020p
 nu}, operating at approximately ten times the energy and baseline of LSND,
 observed an unexpected surplus of electron neutrino and antineutrino events
 in a muon-flavour neutrino beam. These results further highlight the
 unresolved questions surrounding sterile neutrinos and their potential
 implications for neutrino-oscillation phenomena~\cite{Hollenberg:2009tr}.`

“The existence of sterile neutrinos remains controversial, primarily due to the
 challenges in fully understanding systematic uncertainties and backgrounds of
 each experiment. This complexity is reflected in a diverse experimental
 landscape. For instance, measurements sensitive to the same mixing channel
 as KATRIN ($|U_{e4}|^2$), such as Double
 Chooz~\cite{DoubleChooz:2020pnv} and Daya Bay~\cite{DayaBay:2016qvc}
 have reported null results. Moreover, experiments probing complementary
 channels, such as NOvA~\cite{NOvA:2021smv},
 MINOS/MINOS+~\cite{MINOS:2017cae}, and
 Super-Kamiokande/T2K~\cite{T2K:2019efw}, which primarily constrain $\langle |U_{\mu 4}|^2 \rangle$,
 have found no evidence for sterile neutrino mixing. Similarly,
 IceCube/DeepCore~\cite{IceCube:2024dlz}, sensitive to both $\langle |U_{\mu 4}|^2 \rangle$
 and $\langle |U_{\tau 4}|^2 \rangle$, has also reported null results.
 Collectively, these findings reinforce the robustness of the three-flavour
 neutrino framework.”

```

@article{NOvA:2021smv,
  author = "Acero, M. A. and others",
  collaboration = "NOvA",
  title = "{Search for Active-Sterile Antineutrino Mixing Using Neutral-Current  

  Interactions with the NOvA Experiment}",
  eprint = "2106.04673",
  archivePrefix = "arXiv",
  primaryClass = "hep-ex",
  reportNumber = "FERMILAB-PUB-21-271-ND",
  doi = "10.1103/PhysRevLett.127.201801",
  journal = "Phys. Rev. Lett.",
  volume = "127",
  number = "20",
  pages = "201801",
  year = "2021"
}
  
```

```
@article{MINOS:2017cae,  
  author = "Adamson, P. and others",  
  collaboration = "MINOS+",  
  title = "{Search for sterile neutrinos in MINOS and MINOS+ using a  
two-detector fit}",  
  eprint = "1710.06488",  
  archivePrefix = "arXiv",  
  primaryClass = "hep-ex",  
  reportNumber = "FERMILAB-PUB-17-430-ND",  
  doi = "10.1103/PhysRevLett.122.091803",  
  journal = "Phys. Rev. Lett.",  
  volume = "122",  
  number = "9",  
  pages = "091803",  
  year = "2019"  
}
```

```
@article{IceCube:2024dlz,  
  author = "Abbasi, R. and others",  
  collaboration = "IceCube",  
  title = "{Search for a light sterile neutrino with 7.5~years of IceCube  
DeepCore data}",  
  eprint = "2407.01314",  
  archivePrefix = "arXiv",  
  primaryClass = "hep-ex",  
  doi = "10.1103/PhysRevD.110.072007",  
  journal = "Phys. Rev. D",  
  volume = "110",  
  number = "7",  
  pages = "072007",  
  year = "2024"  
}
```

```
@article{T2K:2019efw,  
  author = "Abe, K. and others",  
  collaboration = "T2K",  
  title = "{Search for light sterile neutrinos with the T2K far detector  
Super-Kamiokande at a baseline of 295 km}",  
  eprint = "1902.06529",  
  archivePrefix = "arXiv",  
  primaryClass = "hep-ex",  
  doi = "10.1103/PhysRevD.99.071103",  
  journal = "Phys. Rev. D",  
  volume = "99",
```

```

number = "7",
pages = "071103",
year = "2019"
}

```

- **Question/comment:** *line 193 : $\sin^2(\theta_{ee})$ is used while at line 214 $\sin^2(2\theta_{ee})$ is given. For the casual reader this can be confusing. Maybe a word of explanation would be valuable*

Answer: KATRIN directly measures the mixing parameter $|U_{e4}|^2$, which is equivalent to $\sin^2(\theta_{ee})$. For comparison with oscillation experiments, we also provide the corresponding value of $\sin^2(2\theta_{ee})$. This conversion is explained later in the manuscript (line 462), where we clarify that $\sin^2(2\theta_{ee}) = 4 \cdot \sin^2(\theta_{ee}) \cdot (1 - \sin^2(\theta_{ee}))$. To improve clarity, we modified the previous statement around L193:

“In the context of a $3+1$ neutrino model~\cite{Mention:2011rk}, the extended PMNS matrix U is a 4×4 unitary matrix that describes the mixing between flavour and mass eigenstates. The element $|U_{e4}|^2$, also labeled $\sin^2(\theta_{ee})$, sets the oscillation amplitude~\cite{pdg2023}. For direct comparison with disappearance oscillation experiments, this is often recast as $\sin^2(2\theta_{ee}) = 4 \sin^2(\theta_{ee})(1 - \sin^2(\theta_{ee}))$. A sterile-neutrino signature typically manifests as deviations in the outgoing lepton flux or energy spectrum.”

- **Question/comment:** *line 201: a “few meter” baseline is mentioned and then line 202 a “lower” baseline to 10 to 100 meters is quoted. I find this confusing.*

Answer: We modified the text:

“The gallium anomaly reflects a deficit in electron neutrino flux observed during radiochemical experiments, which involve MeV-scale neutrinos and very short baselines of a few meters. In contrast, the reactor antineutrino anomaly involves discrepancies between predicted and measured fluxes at longer baselines of about 10 to 100 meters, with neutrino energies around 4 MeV.”

Referee 2 and 3

- **Question/comment:** *Some data was excluded due to quality issues (near lines 358 and 385). An additional sentence or two explaining why the quality led to exclusion is warranted.*

Answer: original L358: A total of 1,757 out of 1,895 scans were selected after data-quality cuts, comprising 36 million signal and background electrons within the same region of interest.

We modified L358 as: A total of 1,757 out of 1,895 scans were selected after data-quality cuts, comprising 36 million signal and background electrons within the same region of interest. This selection excludes scans where monitoring systems indicated instabilities in key experimental parameters, such as electromagnetic fields or source conditions.

original L385: Out of the 148 available pixels, 117 were utilized in KNM1 and KNM2, and 126 in KNM3-SAP, KNM3-NAP, KNM4-NOM, KNM4-OPT, and KNM5, with the rest excluded due to quality cuts

We modified L385 as: Out of the 148 available pixels, 117 were utilized in KNM1 and KNM2, and 126 in KNM3-SAP, KNM3-NAP, KNM4-NOM, KNM4-OPT, and KNM5. The remaining pixels were excluded because they either showed increased noise or were partially shadowed by structural components of the beamline.

- **Question/comment:** *In line 215, the word contentious is moderately inflammatory. Replace with the word unconfirmed.*

Answer: original L215: However, these results remain contentious, as no scientific consensus has been reached~\cite{Acero:2022wqg}.

Revised L215: However, these results remain unconfirmed, as no scientific consensus has been reached~\cite{Acero:2022wqg}.

- **Question/comment:** *Line 238: the use of the word “experiment’s” is confusing within the context of the rest of the paragraph. Are the authors referring to the systematic uncertainties of a lone experiment? Or did they mean experiments? If it’s only one experiment, be clear about which one.*

Answer: original L238: The existence of sterile neutrinos remains controversial, primarily because of the difficulties in understanding the experiment’s systematic uncertainties and backgrounds.

Here, taking into account some other referee comments we modified to: “The existence of sterile neutrinos remains controversial, primarily due to the challenges in fully understanding systematic uncertainties and backgrounds of each experiment...”

- **Question/comment:** *Within the paragraph that begins at Line 390, the fit description says that the 3 active squared neutrino mass is fit (line 392), but later it says that parameter is set to zero (line 397). Some editing to make it clear what was done would be useful.*

Answer: original L390-400: In the mass measurement, four key parameters are fitted: the squared active-neutrino mass (m_{ν}^2), the endpoint energy (E_0), the signal amplitude (A_s), and the background rate (R_{bg}). For this test of the sterile-neutrino hypothesis, two additional parameters are introduced: the squared fourth neutrino mass (m_4^2), and the mixing amplitude ($\sin^2(\theta_{ee})$). These parameters are included in the model described in \cref{eq:integralsterilemodeling}. A hierarchical scenario is assumed, where $m_{1,2,3} \ll m_4$. Under this framework, m_{ν} can be set to zero, consistent with the experimental lower limit of (9 meV) for the normal hierarchy ($m_1 < m_2 < m_3$) or (50 meV) for the inverted hierarchy ($m_3 < m_1 < m_2$), derived from neutrino-oscillation data~\cite{pdg2023}, given the sensitivity of the KATRIN measurements.

We revised this paragraph as : “KATRIN measures the effective neutrino mass by analyzing the shape of the tritium β -decay spectrum near the kinematic endpoint. In the standard neutrino mass analysis, four key parameters are fitted: the squared active-neutrino mass (m_{ν}^2), the endpoint energy (E_0), the signal amplitude (A_s), and the background rate (R_{bg}). For the sterile-neutrino analysis, two additional parameters are introduced: the squared mass of the fourth neutrino state (m_4^2) and the mixing amplitude ($\sin^2(\theta_{ee})$). These are included in the model described by equation~(3). Depending on the scenario being tested, the squared active-neutrino mass can either be left as a free parameter in the fit or set to zero. For the main results presented here, we assume a hierarchical scenario where $m_{1,2,3} \ll m_4$, which allows m_{ν} to be set to zero. This is justified by neutrino oscillation data, which constrain the smallest possible effective neutrino mass to about 9 meV for the normal hierarchy ($m_1 < m_2 < m_3$) and 50 meV for the inverted hierarchy ($m_3 < m_1 < m_2$), well below the sensitivity of the current KATRIN measurements.”

- **Question/comment:** *Line 530: The data presented was collected prior to 2021. Providing a sentence regarding what data has been taken since then would be useful in a broader context, especially given the projection section 13.*

Answer: original L530: With the first five measurement campaigns (KNM1-5) conducted between 2019 and 2021, utilizing 36 million electrons near the tritium β -decay endpoint, the KATRIN experiment places new stringent constraints on the existence of light sterile neutrinos.

original L538: With data collection continuing until the end of 2025, KATRIN's sensitivity is expected to improve significantly (see Methods Section 13), offering further insights into the existence of potential light sterile neutrinos.

We modified L538:

With data collection continuing through 2025, KATRIN's sensitivity is expected to increase substantially (see Methods section \cref{sec:meth_forecast}), allowing for improved searches for light sterile neutrinos. The dataset will then exceed 220 million electrons in the region of interest, over six times the current statistics

- **Question/comment:** *The star symbol represented in Fig. 6 should be explained in the caption?*

Answer: We updated the legend

- **Question/comment:** *For the fits, the resulting value of E_0 would be interesting to include and how fixing it to the Penning trap result would, or would not, affect the sterile parameter constraints. Does the uncertainty in E_0 contribute to the systematic uncertainty? Adding the best fit values of E_0 , m_ν , to Table 1 would be nice to see.*

Answer: The effective endpoint energy E_0 represents the Q-value of tritium β -decay (as determined by Penning trap measurements with an uncertainty of 70 meV), modified by work function differences between the source, rear wall, and spectrometer, as well as by the recoil energy of the molecular ion. However, because the absolute electric potential difference between the source and the analyzing plane is only known to within a few hundred millivolts, E_0 cannot be fixed in our fits. As a result, we do not assign a separate systematic uncertainty to E_0 in either the neutrino mass or sterile neutrino analyses, since it is always treated as a free parameter. It is nonetheless important to verify the overall energy scale of the KATRIN experiment by comparing our fitted endpoint with the Q-value from Penning trap measurements. This comparison is performed a posteriori and under a separate blind analysis strategy. For the campaign with the best-controlled source potential, KNM4-NOM, we measure an effective endpoint of (18575.0 ± 0.3) eV, consistent with Penning trap measurements. However, this precision (300 meV) confirms that we need to keep E_0 as a free parameter in

our analyses.

To assess the impact of endpoint treatment on sterile neutrino sensitivity, we performed a dedicated Monte Carlo study for one campaign in which (E_0) was fixed to its nominal value. As expected, this led to improved sensitivity due to the reduced number of free parameters. However, when we instead constrained (E_0) using the Penning trap Q-value uncertainty (i.e., $\sigma_{E_0}=70$ meV), the sensitivity was found to be indistinguishable from the case where (E_0) is left entirely free. This confirms that our current treatment is conservative but robust.

In section “Setup and Dataset” of the main article we added: “The endpoint energy, (E_0), is left free, as it reflects the tritium Q-value and work function differences between the source, rear wall, and spectrometer, introducing a sub-electronvolt uncertainty that prevents it from being fixed in the fit.”

Table of best fit E_0 values (eV):

KNM1: 18573.97 +/- 0.02

KNM2: 18573.67 +/- 0.02

KNM3a: (18573.59 - 18573.77) +/- (0.07 - 0.11), range of 14 patch values

KNM3b: 18573.61 +/- 0.02

KNM4abc: (18573.65 - 18573.77) +/- (0.05 - 0.09), range of 14 patch values

KNM4de: (18573.71 - 18573.81) +/- (0.05 - 0.09), range of 14 patch values

KNM5: (18573.64 - 18573.73) +/- (0.02 - 0.02), range of 14 patch values

- **Question/comment:** *The middle panel of Fig. 4 is very useful --more useful than the panels in Fig. 5. Are there uncertainties on the points in that panel? If so, describe them. If not, consider adding them.*

Answer: The ratio shown in the middle panel of Fig. 4 is purely illustrative. It shows the ratio of simulated spectra in the $3\nu+1$ and 3ν frameworks for a single measurement campaign (KNM5) and one patch (group of pixels with similar electromagnetic fields). This was chosen to highlight the characteristic kink-like feature from a sterile neutrino. As this is a pure simulation, it does not include uncertainties. Adding meaningful error bars is unfortunately not feasible: statistical uncertainties would not be representative of the final results, since the plot reflects only one (over 14) patch and one (over 5) campaign. For reference, the top panel of Fig. 4 does show the KATRIN data with 1σ uncertainties scaled by a factor of 10 for visibility.

- **Question/comment:** *The statistical limit is significantly better than the quoted sensitivity. That implies a statistical fluctuation. In Fig. 4 middle, evidence for a sterile neutrino is a dip in the ratio. For these data, it seems a fluctuation led*

to a bump which provides the improvement over the sensitivity. Perhaps include those plots to give one a feel for any potential fluctuations.

Answer: The ratio shape of 3+1 / 3 neutrino frameworks is dependent on patch-specific and campaign-specific parameters, making it challenging to produce a single representative plot. Furthermore, statistical uncertainties for individual patches are typically too large to draw meaningful conclusions from such residuals given the diversity of measurement conditions across campaigns and patches, it is not feasible to combine all information into a single ratio plot that would summarize residuals versus retarding energy. Instead, the sterile neutrino limits are extracted through a global fit across all campaigns and patches, properly accounting for their individual systematic and statistical contributions.

- **Question/comment:** *The curves representing the systematic uncertainties within the excluded parameter space depicted in Fig. 10 have extended features near 30-50 eV². Can the authors explain the origin of those features?*

Answer: The raster scan contours in Fig. 10 illustrate the impact of individual systematic uncertainties on the sterile neutrino sensitivity. The extended features in the 30-50 eV² sterile mass range indicate that the sterile neutrino sensitivity remains largely unaffected by the inclusion of the individual systematic uncertainties. These features arise because the systematic effects influence the spectrum in a correlated manner, unable to mimic the sterile neutrino signature within these mass ranges. Due to varying correlations for different uncertainties, the feature appears at slightly different m_4^2 values. This feature is also linked to the retarding potential values that we set in our measurement and the distribution of measurement time among them, which affects the sensitivity to sterile neutrinos with specific masses.

- **Question/comment:** *The qualitative conclusions (“excludes much of the parameter space”, “rules out ... with high confidence”) to be a bit strong. The sensitivity is to the right of a large fraction of Neutrino-4 and the key Gallium anomaly space. That is, although the limit excludes much of the sterile neutrino parameter space indicated by previous experiments, the sensitivity does not. With the significant difference between the limit and sensitivity, it is possible that the limit will worsen with further data. The authors should add some discussion to provide a caveat regarding the sensitivity-limit difference and the comparison to other experimental claims.*

Answer: In : “KATRIN excludes much of the parameter space suggested by the reactor and gallium anomalies, except for cases with very small mixing

angles, and rules out the Neutrino-4 claim [7] with high confidence.”, we soften the statements and now have: “KATRIN places stringent constraints on much of the parameter space suggested by the reactor and gallium anomalies, except for cases with very small mixing angles. KATRIN also disfavors the parameter region indicated by the Neutrino-4 claim [7].”

To address the potential impact of statistical fluctuations, we have added a clarification to the discussion of our sensitivity forecasts, noting that future exclusions may also be weaker. This motivation is reflected in the dedicated Methods sections (11 and~13), which describe how such fluctuations are treated and how the projected reach should be interpreted.

We added the following sentence in the manuscript (in Method 13): “Given the observed difference between our current exclusion and the median sensitivity, future data may also yield less stringent limits due to statistical fluctuations. This is accounted for in our sensitivity projections, as shown by the 1σ and 2σ bands in Figure~\ref{fig:final_sensitivity_band}, which illustrate the expected range of statistical variation around the median.”

- **Question/comment:** *There are some citations that should be included or updated. These include:*
 - **Question/comment:** *The SAGE reference [11] is about the solar neutrino results and not the Cr-51/Ar-37 data. Better references are: 51Cr: PRC 59 (1999) 2246, 37Ar: PRC 73 (2006) 045805.*

Answer: We modified the reference to:

```
@article{PhysRevC.59.2246,  
  title = {Measurement of the response of a gallium metal solar neutrino  
  experiment to neutrinos from a  $^{51}\text{Cr}$  source},  
  author = {Abdurashitov, J. N. and Gavrin, V. N. and Girin, S. V. and  
  Gorbachev, V. V. and Ibragimova, T. V. and Kalikhov, A. V. and  
  Khairnasov, N. G. and Knodel, T. V. and Kornoukhov, V. N. and Mirmov,  
  I. N. and Shikhin, A. A. and Veretenkin, E. P. and Vermul, V. M. and  
  Yants, V. E. and Zatsepin, G. T. and Khomyakov, Yu. S. and Zvonarev,  
  A. V. and Bowles, T. J. and Nico, J. S. and Teasdale, W. A. and Wark,  
  D. L. and Cherry, M. L. and Karaulov, V. N. and Levitin, V. L. and Maev,  
  V. I. and Nazarenko, P. I. and Shkol'nik, V. S. and Skorikov, N. V. and  
  Cleveland, B. T. and Daily, T. and Davis, R. and Lande, K. and Lee, C.  
  K. and Wildenhain, P. S. and Elliott, S. R. and Wilkerson, J. F.},  
  collaboration = {The SAGE Collaboration},  
  journal = {Phys. Rev. C},  
  volume = {59},
```

```

issue = {4},
pages = {2246--2263},
numpages = {0},
year = {1999},
month = {Apr},
publisher = {American Physical Society},
doi = {10.1103/PhysRevC.59.2246},
url = {https://link.aps.org/doi/10.1103/PhysRevC.59.2246}
}

```

```

@article{PhysRevC.73.045805,
  title = {Measurement of the response of a Ga solar neutrino
  experiment to neutrinos from a  $^{37}\text{Ar}$  source},
  author = {Abdurashitov, J. N. and Gavrin, V. N. and Girin, S. V. and
  Gorbachev, V. V. and Gurkina, P. P. and Ibragimova, T. V. and Kalikhov,
  A. V. and Khairnasov, N. G. and Knodel, T. V. and Matveev, V. A. and
  Mirmov, I. N. and Shikhin, A. A. and Veretenkin, E. P. and Vermul, V. M.
  and Yants, V. E. and Zatsepin, G. T. and Bowles, T. J. and Elliott, S. R.
  and Teasdale, W. A. and Cleveland, B. T. and Haxton, W. C. and
  Wilkerson, J. F. and Nico, J. S. and Suzuki, A. and Lande, K. and
  Khomyakov, Yu. S. and Poplavsky, V. M. and Popov, V. V. and Mishin,
  O. V. and Petrov, A. N. and Vasiliev, B. A. and Voronov, S. A. and
  Karpenko, A. I. and Maltsev, V. V. and Oshkanov, N. N. and Tuchkov,
  A. M. and Barsanov, V. I. and Janelidze, A. A. and Korenkova, A. V.
  and Kotelnikov, N. A. and Markov, S. Yu. and Selin, V. V. and Shakirov,
  Z. N. and Zamyatina, A. A. and Zlokazov, S. B.},
  journal = {Phys. Rev. C},
  volume = {73},
  issue = {4},
  pages = {045805},
  numpages = {12},
  year = {2006},
  month = {Apr},
  publisher = {American Physical Society},
  doi = {10.1103/PhysRevC.73.045805},
  url = {https://link.aps.org/doi/10.1103/PhysRevC.73.045805}
}

```

- **Question/comment:** *A Recent review article on the gallium anomaly should be cited. Prog. Part. Nucl. Phys 134 (2024) 104082.*

Answer: We added the reference:

```

@article{ELLIOTT2024104082,
title = {The gallium anomaly},
journal = {Progress in Particle and Nuclear Physics},
volume = {134},
pages = {104082},
year = {2024},
issn = {0146-6410},
doi = {https://doi.org/10.1016/j.pnpnp.2023.104082},
url =
{https://www.sciencedirect.com/science/article/pii/S0146641023000637
},
author = {S.R. Elliott and V.N. Gavrin and W.C. Haxton},
keywords = {Solar neutrinos, Electron capture, Radiochemistry,
Oscillations, Sterile neutrinos}}

```

- **Question/comment:** *Ref. 25 is incomplete.*

Answer: We fixed it to:

```

@article{Katrin:2024tvq,
author = {M.~Aker and D.~Batzler and A.~Beglarian and
J.~Behrens and J.~Beisenkötter and M.~Biassoni and B.~Bieringer
and Y.~Biondi and F.~Block and S.~Bobien and M.~Böttcher and
B.~Bornschein and L.~Bornschein and T.~S.~Caldwell and ... (KATRIN
Collaboration)},
title = {Direct neutrino-mass measurement based on 259 days of
KATRIN data},
journal = {Science},
volume = {388},
number = {6743},
pages = {180–185},
year = {2025},
month = apr,
doi = {10.1126/science.adq9592},
eprint = {2406.13516},
archivePrefix = {arXiv},
primaryClass = {nucl-ex}
}

```

- **Question/comment:** *The capitalization of KATRIN is not uniform within the references. (see for example, 23, 42, 44)*

Answer: Fixed.

- **Question/comment:** *A reference to the concept of an Asimov data set should be provided.*

Answer: To add the reference we replaced:

“The process started with the generation of twin Asimov datasets for each measurement campaign, a dataset where all input values are equal to their expected values.” By: “The process began with the generation of twin Asimov datasets for each measurement campaign, where all input values are set to their expected means~\cite{Cowan2011}.”

```
@article{Cowan2011,
  author = {Glen Cowan and Kyle Cranmer and Eilam Gross and Ofer Vitells},
  title = {Asymptotic formulae for likelihood-based tests of new physics},
  journal = {Eur. Phys. J. C},
  volume = {71},
  pages = {1554},
  year = {2011},
  doi = {10.1140/epjc/s10052-011-1554-0},
  eprint = {1007.1727},
  archivePrefix= {arXiv},
  primaryClass = {physics.data-an}
}
```

- **Question/comment:** *The IceCube, MINOS, SuperK, T2K and NOvA collaborations have recently published papers exploring light (around 1 eV² and below) sterile neutrinos. Perhaps these don't fit into the introduction to this work closely enough due to the different mixing channel or because they are null results, but the authors might consider a sentence and appropriate citations.*

Answer: Since the LSND result in 1994 and especially after the combined reactor and gallium anomaly publication in 2011, there have been many important searches for light sterile neutrinos. In our work, we decided to highlight experiments that are primarily sensitive to the channel involving $|U_{e4}|^2$ to keep the discussion focused and easier to follow for readers who are not experts in sterile neutrino physics. We now modified the text and added the following paragraph with additional references to the main article:

“The existence of sterile neutrinos remains controversial, primarily due to the challenges in fully understanding systematic uncertainties and backgrounds of each experiment. This complexity is reflected in a diverse experimental landscape. For instance, measurements sensitive to the same mixing channel as KATRIN ($|U_{e4}|^2$), such as Double Chooz^{\cite{DoubleChooz:2020pny}} and Daya Bay^{\cite{DayaBay:2016qvc}} have reported null results. Moreover, experiments probing complementary channels, such as NOvA^{\cite{NOvA:2021smv}}, MINOS/MINOS+^{\cite{MINOS:2017cae}}, and Super-Kamiokande/T2K^{\cite{T2K:2019efw}}, which primarily constrain $|U_{\mu 4}|^2$, have found no evidence for sterile neutrino mixing. Similarly, IceCube/DeepCore^{\cite{IceCube:2024dlz}}, sensitive to both $|U_{\mu 4}|^2$ and $|U_{\tau 4}|^2$, has also reported null results.

Collectively, these findings reinforce the robustness of the three-flavour neutrino framework.”

Referee 4

- **Question/comment:** *Suggested improvements. No major corrections needed, but two small suggestions:*
 - *a. I question the need for Figure 2 as an illustration in the main text – it doesn’t seem to add much to the discussion, and the method for drawing contours is standard in the field.*

Answer: We acknowledge that, for experts in the field, the method of drawing exclusion contours is indeed standard. However, we would prefer to keep Figure 2 in the main text to assist readers of *Nature* who may not be familiar with our analysis techniques. This figure provides a straightforward visual summary of the statistical approach and helps make the results accessible to a broader audience.

- **Question/comment:** *The color choices in Figure 3 make the different contours (particularly for DANSS, STEREO, and PROSPECT) difficult to interpret. In the legend, GA is usually interpreted to include both GALLEX and SAGE, so BEST+GALLEX+SAGE best fit is somewhat confusing. STEREO and PROSPECT should be in all caps in the legend of the figure, and I recommend including explicit references for each of the curves in the figure that do not come from this work..:*

Answer: We have modified the color scheme in Figure 3 to enhance the visibility of the different contours, making it easier to distinguish experiments.

We also adjusted the legend to use all caps for STEREO and PROSPECT for consistency. We would be happy to finalize the color scheme with the editors based on their experience to further improve readability.

We chose not to include explicit references for each of the curves directly within the figure, as we believe this would crowd the figure too much and reduce its readability. However, all references corresponding to these contours are clearly listed and discussed in the main text.

We kept the RAA curve in the figure, representing it now with a thin dotted line. As the reactor antineutrino anomaly was a major motivation for many sterile neutrino searches, and because no definitive revision of the reactor flux predictions has fully resolved the anomaly, we feel it remains appropriate to show this region for context. We stress in the text that the anomaly could be explained by a bias of reactor neutrino flux predictions.

- **Question/comment:** *Referencing. The null results of sterile neutrino searches with IceCube should also be referenced, for example [Phys. Rev. Lett. 133, 201804 (2024)].*

Answer: We have modified it to incorporate IceCube and other experiments.

Added text “The existence of sterile neutrinos remains controversial, primarily due to the challenges in fully understanding systematic uncertainties and backgrounds of each experiment. This complexity is reflected in a diverse experimental landscape. For instance, measurements sensitive to the same mixing channel as KATRIN ($|U_{e4}|^2$), such as Double Chooz¹ and Daya Bay² have reported null results. Moreover, experiments probing complementary channels, such as NOvA³, MINOS/MINOS+⁴, and Super-Kamiokande/T2K⁵, which primarily constrain $|U_{\mu 4}|^2$, have found no evidence for sterile neutrino mixing. Similarly, IceCube/DeepCore⁶, sensitive to both $|U_{\mu 4}|^2$ and $|U_{\tau 4}|^2$, has also reported null results. Collectively, these findings reinforce the robustness of the three-flavour neutrino framework.”

- **Question/comment:** *Clarity and context. The introduction could be written more cohesively to clearly articulate where KATRIN’s search fits into the landscape of sterile neutrino searches. In many ways, the radioactive source anomalies are the best motivation for KATRIN’s search, and they provide the most direct comparison. The RAA is complicated, and while it is mentioned that the Neutrino-4 result is controversial, the null result of the PROSPECT*

experiment isn't mentioned until 3 paragraphs later. It might be best to discuss each anomaly fully, including null results, in turn in order to give a sufficiently nuanced view of the current state of the field. IceCube results should be included in the discussion of LSND and MiniBooNE.

Answer: We reformulated the introduction as suggested and we have moved the discussion of the PROSPECT null result earlier in the introduction to provide a clearer context alongside the reactor anomaly. Additionally, we have incorporated the IceCube results and other experimental results to present a more comprehensive overview of the current state of sterile neutrino searches.

Referee 5

- **Question/comment:** *I think that assuming the active neutrino mass to be zero for extracting a limit on the sterile neutrino mass is not logically consistent. KATRIN is an experiment to measure neutrino masses and arguably the most sensitive one, therefore, all information on neutrino masses used in the fit should come from its own data. This is for instance the approach taken in the KATRIN main result paper on the active neutrino mass, where negative mass squared values are allowed in the fit. Indeed, KATRIN is the first tritium experiment to find mass squared values that are consistent with being zero, see Fig 3 of <https://doi.org/10.1126/science.adq9592>. Therefore, the dashed yellow line (m_n^2 free) of Fig 15 in the current manuscript should be the main limit shown in Fig 3. This is important because KATRIN is the most direct, i.e. least model-dependent, test of the sterile neutrino hypothesis. Adding the constraint on the active neutrino mass squared makes this result less valuable.*

Answer: In the KATRIN neutrino mass measurement, we allow the squared neutrino mass parameter to float freely, including negative values, precisely as the referee points out. In that analysis, the best fit yields $m\nu^2 = -0.14 \pm 0.15$ eV², showing a slight negative value well within uncertainties. This is appropriate for assessing the absolute neutrino mass scale in a model-independent way.

The situation in the sterile neutrino search is structurally different. Here, the tritium beta decay spectrum is modeled by two distinct branches: one associated with the active neutrino states, and another with the hypothetical sterile state, as shown in Eq. (2) of the manuscript. In this case, the data becomes sensitive not only to the individual branch but also to their interplay. If both the active and sterile masses are left completely unconstrained, large sterile-to-active mixing can cause the two branches to swap roles or compensate each other to reproduce the data. This may drive the active mass

squared to large negative values (an “overshoot”) balanced by large positive sterile masses. Such solutions are structurally possible in the fit but are likely unphysical. To prevent this, we apply a technical constraint ensuring the hierarchy $0 < m_\nu < m_4$, which avoids these degenerate or compensating scenarios and maintains the physical interpretation of the two branches. This interplay and the necessity of this approach were discussed in detail in our previous sterile neutrino analysis, Phys. Rev. D 105 (2022) 7, 072004, Section VI.

We believe that presenting the results with $m_\nu^2 = 0$ as our main exclusion contour remains the most appropriate choice for this sterile neutrino analysis, for the following reasons:

- If m_ν is left free, then at each point on the sterile parameter grid (m_4^2 , $\sin^2\theta_{ee}$) the fit can yield a different value for m_ν , and therefore for the masses m_1 , m_2 , m_3 . This means we cannot consistently construct Δm_{41}^2 to compare our results with neutrino oscillation experiments. Fixing $m_\nu^2 = 0$ ensures a clear and consistent mapping in the standard oscillation parameter space.
- We have also shown (in Methods Section 12 and Fig. 15) that applying a minimal technical constraint, $0 < m_\nu^2 \leq m_4^2$, leads to results that are essentially the same as fixing $m_\nu^2 = 0$.

In addition, we provide the results with fully free active neutrino mass in Methods Section 12 and Fig. 14, together with the values of the neutrino masses fitted as color-coded. We clearly see values well above and below 0.15 eV^2 which are inconsistent with the neutrino mass analysis and arise because of the artificial compensation of the sterile and active neutrino branches.

- **Question/comment:** *In terms of framing this result as addressing the so called reactor antineutrino anomaly (RAA), I would like to stress that there is an emerging consensus among experts that the RAA is not due to a sterile neutrino but due to a normalization problem with the beta spectrum data collected in the 1980/90s at the Institut Laue-Langevin. The introduction contains some hedging phrases acknowledging this, but the figures prominently display the RAA allowed region. So I would like to suggest removing the RAA region from Figure 3.*

Answer: We fully agree that there is growing evidence that the reactor antineutrino anomaly may be explained by biases in the reactor flux predictions, rather than by sterile neutrinos. We already highlight this interpretation in the text. However, since the RAA played a central role in motivating many of the sterile neutrino searches over the past decade, we believe it is still valuable to include this region in Figure 3 for historical and contextual perspective. To ensure it does not dominate the figure or mislead

the reader, we now show the RAA contour as a thin dotted line, visually distinguishing it from more significant and robust GA.

Additional data/code

- **Question/comment:** Referee #2: *The editor mentioned there was going to be a codebase/dataset made available to the community as part of this paper and asked for an assessment of usability by the reader. However, the Zenodo link that was sent as follow-up was specifically for the previous mainline KATRIN result for the direct neutrino mass measurement and not this specific sterile neutrino search (<https://zenodo.org/records/13644900>). This analysis is complex and requires large scale computing resources to calculate the various final exclusion curves therefore data sharing is complicated. Even so, it would help the community to release some part of the final high-level analysis/sensitivity (final spectra and exclusion curves, e.g.) as an open-source dataset with instructions included with the raw data. The Zenodo link only contains the raw json files and requires that the user go to the last section of the mainline KATRIN paper to find an explanation. The open source data releases of the CDMS WIMP dark matter search experiment could serve as an example: <https://supercdms.slac.stanford.edu/science-results/data-releases>. While CDMS also has large scale data sets with multiple cuts, most of these data releases allow others to download and reproduce most of the final data curves that go into calculating the final sensitivity limit. At the same time something that is vital for these data releases is the accompanying documentation/explanations again see CDMS data releases that allow for other members of the community to compare their calculations with the potential sensitivity limits. spanning the range of parameters in Fig 3. That at least would ensure proper inclusion in global fits. Further, maybe it is conceivable to augment the data release with a simple χ^2 -model including the leading systematics (maybe in the form of a covariance matrix)? This would allow the wider pheno community to test a range of new physics scenarios.*
- **Question/comment:** Referee #3: *The data release on <https://doi.org/10.5281/zenodo.13644899> is somewhat minimal and it's unclear if anyone would be able to reproduce Fig 3 from that data. I would like to suggest in addition to also release a χ^2 table spanning the range of parameters in Fig 3. That at least would ensure proper inclusion in global fits. Further, maybe it is conceivable to augment the data release with a simple χ^2 -model including the leading systematics (maybe in the form of a covariance matrix)? This would allow the wider pheno community to test a range of new physics scenarios.*

Answer: Thank you for your helpful suggestions about data availability and reproducibility. We agree that transparent data sharing is important so others can check our results and use them in publications. To support this, we have extended our data release. We now provide:

1. The data underlying each figure from this analysis in JSON formats.
2. For the contour plots in particular, we include the 95% exclusions contour in the $(\sin^2(\theta_{ee}), \Delta m^2)$ plane, along with the global minimum chi-square and the best-fit parameters.
3. Figures including pure simulation information are not provided in the data release (Figure 2, Figure 4, Figure 12)
4. Figure 5 data has already been released along with the neutrino mass article at <https://zenodo.org/records/13644900>.

These data sets are now hosted in a dedicated *KATRIN Sterile Neutrino Analysis* repository linked to our Zenodo community at <https://zenodo.org/records/15860995>.

At this stage, the full KATRIN analysis software is not publicly available. The codebase is extensive, has evolved over many years, and relies on multiple interdependent components developed across the collaboration. All computations presented here are based on well-documented frameworks described in previous publications, including the SSC~\cite{Kleesiek:2018mel} and Netrium~\cite{Karl:2022pli} packages. While a public release of the full software stack is conceivable in the future, it would require substantial coordination and validation efforts that extend beyond the scope of this study. Moreover, reproducing the sterile neutrino analysis would not be trivial, as it involves large-scale simulations requiring approximately 50,000 CPU days. The software and analysis infrastructure also remain in active use for ongoing KATRIN searches, including tests of non-standard neutrino interactions and relic neutrino signatures.

Sterile-neutrino search based on 259 days of KATRIN data: Answer to Referee Comments

Thierry Lasserre

1 Answer to Referee #2

Below is the answer to the comment of referees #2 and #3

We thank the referees for this useful suggestion. In response, we have added a README file to the Zenodo repository. Please note that we have released a new version of our data repository (version 2), which now includes this README file with explanations of the JSON files and additional commentary for the phenomenology community.

The data and analysis inputs as well as the results have been deposited at Zenodo (<https://zenodo.org/records/15860995>, <https://doi.org/10.5281/zenodo.15860994>).

2 Answer to Referee #5

Below is the answer to the comment of referee #5

We agree with the referee's and therefore we provide results, with and without constraints on the active neutrino mass completely (see references to Methods Section 12). At the same time, we would like our results to be presented in a way that our main summary plot to be directly comparable to other sterile-neutrino searches, mainly carried out by oscillation experiments. For this purpose, the fourth-neutrino mass (y-axis) must be expressed as a squared mass splitting between the fourth and first neutrino states. To perform this conversion without ambiguity, the active neutrino mass cannot be left floating. We therefore fix the active neutrino mass to zero.

We noticed that in the main text of the paper we write: "Figure 3 assumes $0 \leq m_\nu^2 < m_4^2$, ensuring that m_ν^2 remains positive and below the sterile-neutrino mass squared. Alternative assumptions, including those with free m_ν^2 , are discussed in Methods Section 12." This wording could be misleading. We therefore rephrased it as: "Figure 3 is obtained under the assumption $m_\nu^2 = 0$, so that the sterile-neutrino mass parameter corresponds directly to the squared mass splitting relative to the active state. Alternative treatments, including cases where m_ν^2 is left free, are discussed in Methods Section 12."

We fully agree that in the neutrino mass analysis, allowing the neutrino mass squared to fluctuate below zero is essential, since it provides a check for possible unaccounted systematic effects, as was historically the case in earlier β -decay experiments. At present, KATRIN finds a best-fit neutrino mass squared that is slightly negative, but this is fully consistent with a statistical fluctuation within one standard deviation. This justifies considering the case of zero active neutrino mass when presenting sterile-neutrino results in Fig. 3. We also emphasize that both the neutrino mass and sterile-neutrino analyses are currently statistically limited.

We thank the referee for the careful reading and the valuable discussions. We believe we now provide all results and data needed for experts to fully assess our findings. For the main presentation, we choose to fix the active neutrino mass to zero, as this allows a direct comparison with oscillation experiments. The alternative cases are also included (Methods Section 12, Fig. 15) and referenced in the main text. Finally, we follow the tradition of our previous publications in displaying the results in this way. We therefore believe this approach is the most suitable for a broad audience, while keeping the full technical information available in the manuscript.